# WOW-SEG: A WORD-FREE OPEN WORLD SEGMENTATION MODEL

**Danyang Li**[2,1,3]* **Tianhao Wu**[4]* **Bin Lin**[5] **Zhenyuan Chen**[2,1,3] **Yang Zhang**[2,1,3]
**Yuxuan Li**[2,1,3] **Ming-Ming Cheng**[1,2,3] **Xiang Li**[1,2,3]†

[1]NKIARI, Shenzhen Futian [2]VCIP, CS, Nankai University [3]AAIS, Nankai University
[4]Sichuan Agricultural University [5]Peking University Shenzhen Graduate School

{danyang.li,xiang.li.implus}@nankai.edu.cn

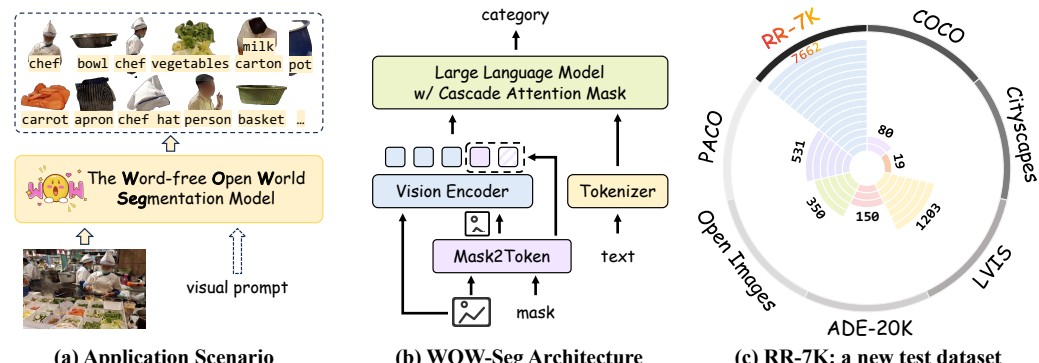

(a) Application Scenario  (b) WOW-Seg Architecture  (c) RR-7K: a new test dataset

Figure 1: (a) Application scenarios for WOW-Seg. WOW-Seg accepts visual prompts of any form as input, outputting corresponding masks and categories. (b) Core architecture of WOW-Seg: Mask2Token and Cascade Attention Mask. Mask2Token converts input masks into visual tokens aligned with the VLLM feature space. Cascade Attention Mask addresses the issue of "inter-instance interference" by decoupling multi-mask features, enabling efficient parallel training. (c) Comparison of the proposed Region Recognition Dataset (RR-7K) with other datasets in terms of categories.

## ABSTRACT

Open world image segmentation aims to achieve precise segmentation and semantic understanding of targets within images by addressing the infinitely open set of object categories encountered in the real world. However, traditional closed-set segmentation approaches struggle to adapt to complex open world scenarios, while foundation segmentation models such as SAM exhibit notable discrepancies between their strong segmentation capabilities and relatively weaker semantic understanding. To bridge these discrepancies, we propose WOW-Seg, a Word-free Open World Segmentation model for segmenting and recognizing objects from open-set categories. Specifically, WOW-Seg introduces a novel visual prompt module, Mask2Token, which transforms image masks into visual tokens and ensures their alignment with the VLLM feature space. Moreover, we introduce the Cascade Attention Mask to decouple information across different instances. This approach mitigates inter-instance interference, leading to a significant improvement in model performance. We further construct an open world region recognition test benchmark: the Region Recognition Dataset (RR-7K). With 7,662 classes, it represents the most extensive category-rich region recognition dataset to date. WOW-Seg attains strong results on the LVIS dataset, achieving a semantic similarity of 89.7 and a semantic IoU of 82.4.

---

*Equal Contribution
†Corresponding Author

This performance surpasses the previous SOTA while using only one-eighth the parameter count. These results underscore the strong open world generalization capabilities of WOW-Seg. The code and related resources are available at `https://github.com/AAwcAA/WOW-Seg-Meta`.

# 1 INTRODUCTION

As a fundamental technology in computer vision, image segmentation aims to deconstruct images into regional units with semantic value. This technology is widely applied in autonomous driving (Muhammad et al., 2022; Zendel et al., 2022), medical image analysis (Azad et al., 2024; Butoi et al., 2023), image editing and synthesis Chen et al. (2019); Meng et al. (2025), among other fields. Throughout the development of image segmentation research, technological iterations have largely centred on optimising accuracy and enhancing efficiency (Ronneberger et al., 2015; Guo et al., 2022; Xie et al., 2021; Cheng et al., 2022). However, when confronted with the boundless variety of object categories and complex, ever-changing scene combinations in the real world, these methods commonly suffer from reliance on closed datasets and limitations in model transferability (Li et al., 2022). They struggle to satisfy segmentation demands in dynamic scenarios, necessitating breakthrough solutions to propel progress in the field.

Deep learning based image segmentation algorithms can be categorized into three types: closed-set segmentation, open-vocabulary segmentation, and vision language model based segmentation. Closed-set methods assign each pixel to predefined categories via a fixed segmentation head (Ronneberger et al., 2015; Guo et al., 2022; Xie et al., 2021; Cheng et al., 2022), while open-vocabulary methods match pixel regions with category labels, though their openness is limited by vocabulary size (Li et al., 2022; Rao et al., 2022; Dong et al., 2023). Vision language models drive segmentation through text instructions, but results heavily depend on textual guidance (Lai et al., 2024; Yang et al., 2023; Wei et al., 2025; Ren et al., 2024b). Consequently, these methods are constrained in the categories they can segment and struggle in open-world scenarios.

Recent advances explore open-world segmentation: SAM (Kirillov et al., 2023) and SAM 2 (Ravi et al., 2024) segment objects class-agnostically, but lack regional semantic understanding. DAM incorporated positional information into the Localised Vision Backbone to enable regional understanding (Lian et al., 2025), and PAM combined SAM2 with large language models for diverse regional semantic outputs (Lin et al., 2025). However, both only handle a single mask during training and inference. VP-MLLM introduced a visual prompt encoder and multi-mask training, yet ignored interference between correlated masks (Lin et al., 2024).

Our analysis reveals that the development of current open world segmentation models is constrained by several key factors. Firstly, both existing close set segmentation methods and open vocabulary segmentation approaches are limited in their output capabilities by predefined categories. Secondly, segmentation approaches based on visual large language models necessitate the appropriate incorporation of visual prompts. Moreover, prior open world segmentation models can only process a single mask during a single forward inference, resulting in speed limitations in multi instance interaction scenarios (Lian et al., 2025; Lin et al., 2025; Yuan et al., 2024). Additionally, most existing segmentation datasets contain only tens to a thousand categories, failing to comprehensively evaluate a model's open world understanding capabilities. To address the current limitations of segmentation models in open world scenarios, we propose WOW-Seg.

As shown in Fig. 1(a), WOW-Seg is a word-free open world segmentation model. It segments open world scenes through visual prompts and uses the powerful cognitive abilities of VLM to recognize masks. The core architecture of WOW-Seg is shown in Fig. 1(b). We have devised a novel visual prompting approach: Mask2Token. It extracts features by processing specific masked regions, ensuring the mask tokens (purple squares) align with the VLLM's feature space while pruning irrelevant background tokens (purple chequered squares). Moreover, our proposed Cascade Attention Mask mitigates interference between object instances by decoupling their respective features during multi-instance training and inference. To comprehensively evaluate the performance of open world segmentation models, we propose a new test benchmark: the Region Recognition (RR-7K). Its comparison with other datasets is shown in Fig. 1(c). RR-7K comprises 7,662 categories, over 80k images, and more than 200k mask objects. Compared to previous evaluation methods using

datasets such as LVIS, RR-7K features richer categories, enabling a more thorough assessment of models' understanding capabilities in open world scenarios.

Overall, our contributions are as follows:

- WOW-Seg is a word-free open world segmentation model that encodes multiple masks as visual prompts with Mask2Token and autoregressively identifies each mask via Cascade Attention Mask, enabling detection of any object without text guidance.

- We introduce RR-7K, an open-world region recognition dataset with 7,662 categories and more than 200k masks, providing a comprehensive benchmark for evaluating model generalization.

- WOW-Seg achieves state-of-the-art performance with fewer parameters, surpassing previous SOTA by 4.1 Semantic IoU on LVIS and 4.3 on PACO.

## 2 RELATED WORK

**Open-set image segmentation.** Closed-set image segmentation algorithms have achieved remarkable performance. However, they rely on predefined categories, which limits their ability to handle novel classes or adapt to new domains (Bucher et al., 2019; Yin et al., 2024). Open-vocabulary segmentation extends semantic coverage by integrating text embeddings with visual features (Dong et al., 2023; Li et al., 2022). Rao et al. (2022); Chen et al. (2024b) reformulates CLIP's image-text matching as a pixel-level task for dense predictions. Despite these advances, open-vocabulary methods still require predefined categories and cannot predict truly unknown concepts. The Segment Anything Model achieves zero-shot segmentation without category annotations (Kirillov et al., 2023; Ravi et al., 2024), but cannot assign semantic meaning to the segmented regions. To reduce reliance on predefined categories, some studies explore unguided and automatic vocabulary models (Kawano & Aoki, 2024; Ülger et al., 2025; Rewatbowornwong et al., 2023; Shin et al., 2024), attempting to autonomously discover semantic labels. In contrast, our method encodes multiple masks as visual prompts and employs an autoregressive approach to identify each mask independently, allowing detection of any object without text guidance.

**Vision Language Large Model.** Vision-language large models (VLLMs) exhibit strong generalization in tasks such as image captioning and visual question answering by jointly learning visual and textual representations (Liu et al., 2023; Chen et al., 2024a; Bai et al., 2025; Shabbir et al., 2025; Liu et al., 2025; Li et al., 2025; Zhu et al., 2025; Zhang et al., 2025; 2026; Zhao et al., 2025b;a). To transfer this world-level cognitive ability to segmentation, Lai et al. (2024) proposed LISA, which segments user-specified objects based on text instructions. However, its performance remains heavily dependent on user-provided text. Efforts to incorporate regional understanding into VLLMs have introduced additional encoders to embed both image features and positional information of masked regions (Lin et al., 2024; Yuan et al., 2024; Rasheed et al., 2024). Yet, the tokens produced by these encoders lie outside the original VLLM feature distribution, requiring extensive training for alignment. Moreover, (Lin et al., 2025; Lian et al., 2025; Yuan et al., 2024) rely on single-mask samples during training and inference, substantially prolonging model training. While VP-MLLM (Lin et al., 2024) supports reasoning over multiple masks in a single sample, it does not account for inter-mask correlations, and direct multi-mask training leads to interference that degrades performance. In contrast, our approach leverages the Cascade Attention Mask module to predict each mask independently, avoiding semantic interference between correlated masks and improving the efficiency of multi-mask parallel inference.

## 3 METHOD

To achieve the objective of segmenting open world scenes using solely visual prompts, we designed WOW-Seg, an innovative segmentation and recognition framework based on autoregressive mechanisms. The core concept of WOW-Seg is to transform the object category recognition task from a traditional classification problem into a visually-driven text generation problem. The entire framework is shown in Fig. 2. It consists of four core modules: Mask2Token, Cascade Attention Mask, Mask Generator and VLLM.

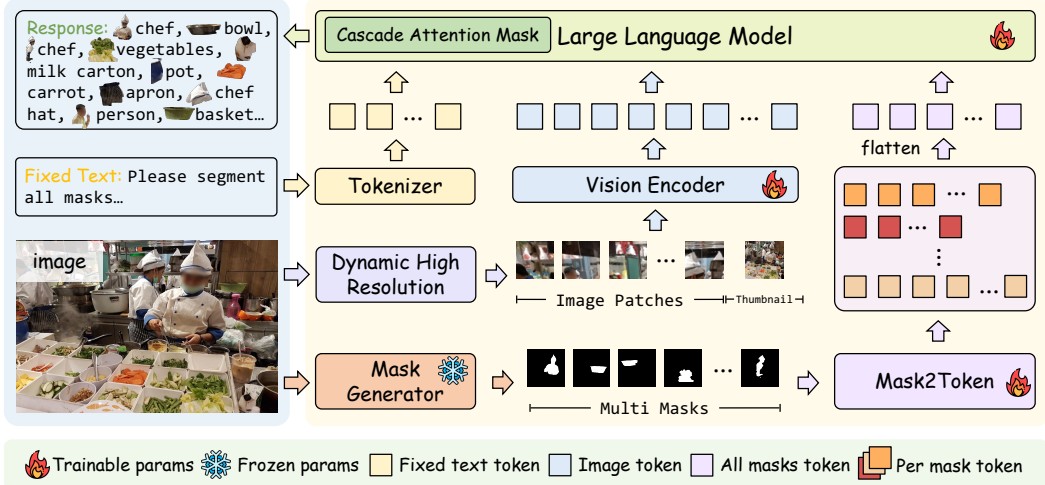

Figure 2: The overall framework of WOW-Seg. WOW-Seg will perform corresponding tokenisation on fixed text and input images. WOW-Seg employs Mask2Token to convert the received mask into visual tokens. These tokens are then collectively fed into a large language model featuring a Cascade Attention Mask mechanism. Ultimately, WOW-Seg completes the open-world category output.

## 3.1 WOW-SEG FRAMEWORK

The proposed WOW-Seg model is built upon an encoder-decoder framework and utilizes a VLLM as its foundation. Given an image and a set of corresponding masks as input, the model autoregressively generates the category name for each mask. The visual encoder first processes the input image to extract a sequence of image tokens, which provide rich contextual information for recognition. During the training phase, we use ground truth masks as input. During the inference phase, users may flexibly select from multiple mask generators to produce masks, such as SAM (Kirillov et al., 2023). The Mask2Token module processes the input masks by mapping them into tokens within the Vision Language Model's (VLM) embedding space. These mask tokens, along with tokens from the image and a predefined text prompt, are then fed into the Large Language Model (LLM) decoder. Within the decoder, we employ our novel Cascade Attention Mask to ensure that the prediction for each mask is generated independently, mitigating interference between instances. By leveraging the standard "next token prediction" mechanism of the LLM, our approach bypasses the fixed-category limitations of traditional segmentation heads, enabling the model to recognize an open set of categories.

## 3.2 MASK2TOKEN

Prior methods have often employed a dedicated module to encode binary masks into feature tokens for downstream Vision Language Models (Lin et al., 2024). While this enables the integration of mask information, the resulting tokens introduce a distributional gap, as they exist outside the VLM's pretrained embedding space. Bridging this gap typically requires substantial retraining to align the new feature distribution.

The Mask2Token module, illustrated in Fig. 3, is designed to embed mask-specific visual and spatial features into the VLM. For each input mask, the module first crops a mask region image that includes the mask and its surrounding context (default context scale: 2). The mask region image is then resized to the visual encoder's standard input dimension ($448 \times 448$) and processed by a shared-weight vision encoder to produce a grid of $16 \times 16$ image tokens. Using a shared-weight encoder is crucial as it ensures the resulting features lie in the same embedding space as the global image features. Concurrently, the binary mask is downsampled to a $16 \times 16$ resolution. Finally, this downsampled mask is used as a guide to select the corresponding tokens from the feature grid. Notably, Mask2Token processes multiple masks in parallel, enabling features from several distinct objects to be fed into the LLM simultaneously.

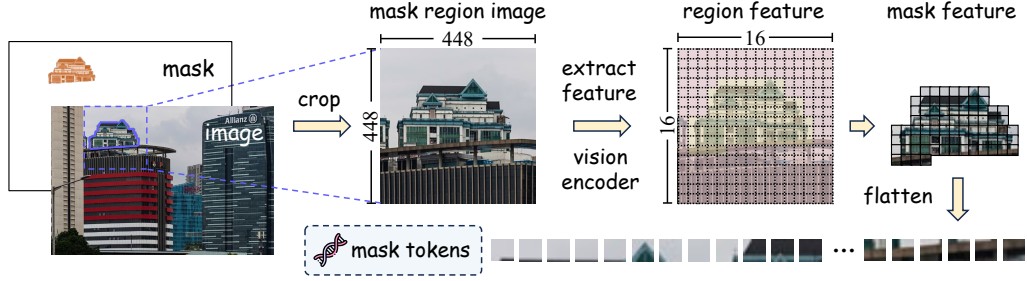

Figure 3: Detailed illustration of our Mask2Token. Only the processing procedure of a mask is presented here. In fact, Mask2Token receives multiple masks and performs parallel processing.

## 3.3 CASCADE ATTENTION MASK

While a Vision-Language Model can be trained using a single mask per sample (SM), this approach is less efficient than training with multiple masks per sample (MM). The MM strategy allows the model to process all object instances in an image within a single forward pass, which better reflects the nature of open-world scenes that typically contain multiple independent objects. However, this MM strategy introduces a critical challenge: inter-instance interference, where the model may incorrectly associate features from different objects. To explicitly prevent this detrimental information leakage, we designed the Cascade Attention Mask.

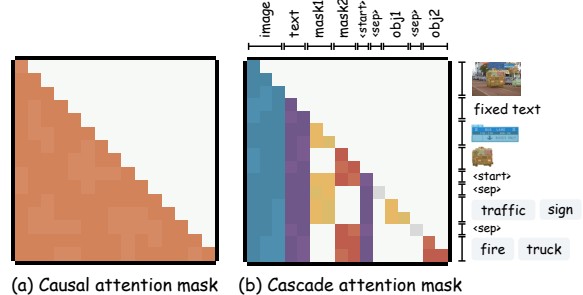

Figure 4: Causal Attention Mask and Cascade Attention Mask.

Under the effect of the original causal attention mask of the large language model, assuming there are $K$ objects, the probability of these $K$ object names being predicted is shown in the following formula:

$$P\left(O_1, O_2, \ldots, O_K \mid Image; T; M\right) = \prod_{i=1}^{K} P\left(O_i \mid Image; T; M; O_0, O_1, \ldots, O_{i-1}\right). \quad (1)$$

Among them, $O$ represents the object to be predicted, $Image$ represents the image token, $T$ represents the text token, and $M = \{m_1, m_2, \ldots, m_K\}$ represents the mask tokens. It is easy to observe that when predicting the $i$-th object, the model refers to the mask prompts and output results from the 0-th to the $(i-1)$-th objects. We hope that the name of the $i$-th object is guided solely by its corresponding $i$-th mask. Therefore, we designed the Cascade Attention Mask.

The detailed mechanism of the Cascade Attention Mask is illustrated in Fig. 4(b), which we explain using an example of a single image containing two masks: a "traffic sign" and a "fire truck". The image token corresponds to the blue region. The text token corresponds to the purple region, with the input text $\langle start \rangle$ also corresponding to the purple region. The features corresponding to the first object (traffic sign) are represented by the yellow region. The features corresponding to the second object (fire truck) are represented by the red region. $\langle sep \rangle$ serves as the separator. The attention is structured to ensure their category names are decoded independently. Tokens corresponding to the image and the text prompt are universally accessible to all other tokens. However, the core of our approach lies in isolating the instance-specific tokens. The input mask tokens for "traffic sign" and "fire truck" are masked from each other, preventing information leakage. During autoregressive generation, the process is as follows: to generate the output "traffic sign", the model can only attend to the input mask tokens for the traffic sign and its own previously generated tokens (e.g., "traffic"). Crucially, it cannot attend to the mask tokens of the "fire truck". Similarly, the generation of "fire truck" is causally conditioned only on its own mask tokens and preceding outputs. This parallel yet

decoupled decoding process ensures that the prediction for each object is not influenced by others. Overall, the design adheres to the following principles:

**1. Masks are independent of each other.** Masks appearing later do not carry any information from earlier masks.

**2. Objects are independent of each other.** The prediction of object $i$ is not affected by objects 1 to $i-1$.

**3. Decouple each pair of mask and object.** When predicting object $i$, the information available to the model is limited to image tokens, text prompt tokens, and the $i$-th mask token.

When predicting the object $i$, the only information that the model can obtain are image tokens, text tokens, and the i-th mask token. Ultimately, the Cascade Attention Mask enables the model to learn without interference from additional factors. When designing the Cascade Attention Mask, we also explored its variants. The specific design of the variant is detailed in Fig. 10 of the appendix. Its primary distinction from the Cascade Attention Mask lies in whether it simultaneously adheres to the principles that "Masks are independent of each other" and "Objects are independent of each other". We also conducted ablation experiments comparing the performance of the Cascade Attention Mask and its variants, as shown in Tab. 4.

Under the Cascade Attention Mask, the predictions for different objects are conditionally independent. Specifically,

$$P\left(O_1, O_2, \ldots, O_K \mid Image; T; M\right) = \prod_{i=1}^{K} P\left(O_i \mid Image; T; M\right), \tag{2}$$

where $M = \{m_1, m_2, \ldots, m_K\}$ denotes the set of all object mask tokens. This factorization reflects the fact that, under the Cascade Attention Mask, the prediction of the $i$-th object $O_i$ is conditionally independent of the mask tokens of all other objects given the image and text tokens. Although the conditioning set $M$ includes the mask tokens of all objects, the attention pattern enforces that $O_i$ can only attend to its own mask tokens $m_i$. As a result, the probability term satisfies:

$$P\left(O_i \mid Image; T; M\right) = P\left(O_i \mid Image; T; m_i\right), \tag{3}$$

meaning that only the $i$-th mask contributes to the prediction of the $i$-th object.

## 3.4 RR-7K DATASET

Currently, the performance evaluation of open world models is mainly conducted on common classes. To comprehensively measure the performance of open-world models, we propose the Region Recognition Dataset (RR-7K).

The images and mask annotations of RR-7K are from SA-1B (Kirillov et al., 2023). We designed a sophisticated data annotation pipeline to assign categories to each mask in RR-7K. As shown in Fig. 5, the pipeline consists of three stages.

**Mask patch category inference.** There are a large number of meaningless tiny masks in SA-1B, which greatly increases the inference cost but fails to obtain high-quality data. Therefore, we remove masks where the proportion of mask pixels relative to the total pixels of the entire image is excessively low. Subsequently, for each mask, we used tools such as Qwen2.5VL-72B (Bai et al., 2025) and Grounded SAM (Ren et al., 2024a) to obtain the corresponding category.

**Hallucination filtering.** After the category inference of mask patches, we obtained a large number of mask-category pairs. These pairs exhibit a long tailed distribution, and there is a large amount of incorrect data caused by the hallucination phenomenon of large language models. For tail categories, the cognitive ability of large language models is not good. Therefore, we use InternVL-78B to perform hallucination filtering only on the data of head categories. We will re-ask a question about a mask-category pair: *"Is the area outlined by the red contour and covered by the red mask in the image {class name}? Please answer yes or no."* By re-asking questions for each mask-category pair, a large amount of incorrect data was screened out.

**Manual screening.** We will manually screen all the tail categories and the head categories after hallucination filtering. Since each mask now has a category name, we only need to delete the samples with inconsistent categories, which incurs low labor costs.

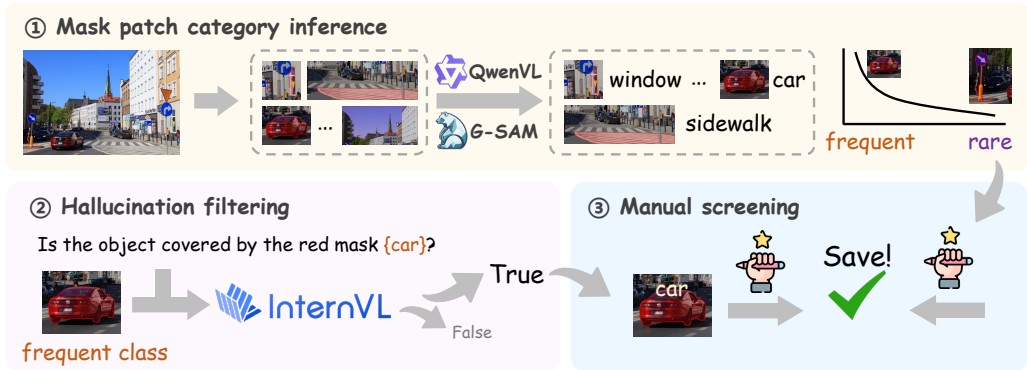

Figure 5: Data annotation pipeline. The RR-7K we proposed mainly goes through three stages: Mask patch category inference, Hallucination filtering and Manual screening.

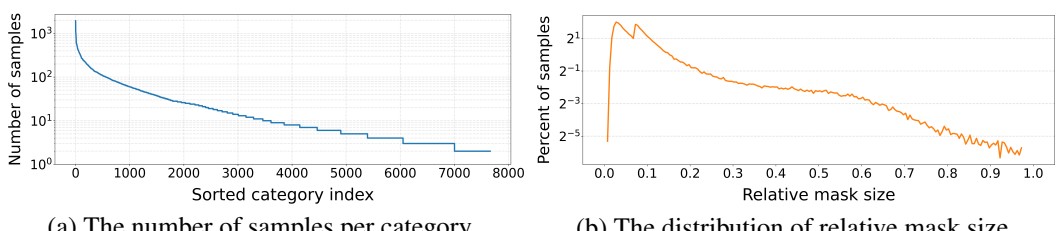

(a) The number of samples per category.   (b) The distribution of relative mask size

Figure 6: Statistical distribution of RR-7K. The relative mask size denotes the square root of the mask area divided by the image area.

After going through the above data annotation pipeline, we obtained the final RR-7K. It contains over 80k images, with more than 200k instances and 7,662 categories. The distribution of sample counts per category and the relative size distribution of masks are shown in Fig. 6(a) and Fig. 6(b) respectively. Further details are presented in the appendix. The RR-7K contains a large number of categories in the open world, providing a better test benchmark for open world segmentation and understanding.

## 4 EXPERIMENTS

### 4.1 IMPLEMENTATION DETAILS

We use the pre-trained InternVL3-1B (Zhu et al., 2025) as the base model for WOW-Seg. The model is trained on 8 NVIDIA H100 GPUs. We use the AdamW optimizer with a learning rate set to $1e - 5$ and a batch size set to 32. For the final reported model performance, we use LVIS (Gupta et al., 2019), PACO (Ramanathan et al., 2023), and COCO Stuff (Caesar et al., 2018) for 2 epochs of training. By default, the maximum number of masks contained in a single data sample used in all experiments does not exceed 30. This is to ensure load balancing on each GPU during multi-GPU training.

### 4.2 OPEN WORLD REGION RECOGNITION

This task requires identifying the object category within a masked area without a given vocabulary or any other text prompts. We follow the settings of Yuan et al. (2024) and Lin et al. (2024), and use semantic similarity and semantic intersection over union to measure the performance of the model. We evaluate the model on three datasets in total. LVIS (Gupta et al., 2019), a large-scale instance segmentation benchmark, contains over 1,200 long-tailed distributed categories, aiming to test the model's generalization ability to rare objects. PACO (Ramanathan et al., 2023), a dataset focusing on fine-grained understanding. It not only contains complete objects but also provides detailed

Table 1: Results of open world region level image recognition on LVIS, PACO and RR-7K.

| Model | Params | LVIS | | PACO | | RR-7K | |
|---|---|---|---|---|---|---|---|
| | | Semantic Similarity | Semantic IoU | Semantic Similarity | Semantic IoU | Semantic Similarity | Semantic IoU |
| Shikra (2023) | 7B | 49.7 | 19.8 | 43.6 | 11.4 | - | - |
| GPT4RoI (2024) | 7B | 51.3 | 12.0 | 48.0 | 12.1 | - | - |
| Osprey (2024) | 7B | 65.2 | 38.2 | 73.1 | 52.7 | 61.0 | 32.5 |
| Ferret (2023) | 13B | 65.0 | 37.8 | - | - | - | - |
| VP-LLAVA (2024) | 8B | 86.7 | 61.5 | 75.7 | 50.0 | - | - |
| VP-SPHINX (2024) | 13B | 87.1 | 62.9 | 76.8 | 51.3 | 54.7 | 17.5 |
| DAM (2025) | 8B | 89.0 | 77.7 | 84.2 | 73.2 | - | - |
| PAM (2025) | 1.5B | 87.4 | 76.5 | 85.1 | 73.5 | 44.5 | 12.4 |
| PAM (2025) | 3B | 88.6 | 78.3 | 87.4 | 74.9 | 45.8 | 13.4 |
| **WOW-Seg (Ours)** | **1B** | **89.7** | **82.4** | **88.5** | **79.2** | **69.1** | **44.8** |

annotations for their components, posing higher requirements for the model's part perception ability. RR-7K, the open world region recognition test benchmark proposed in this paper, contains 7,662 categories and aims to test the model's understanding of the open world.

As shown in Tab. 1. WOW-Seg achieves the best performance on LVIS, PACO, and RR-7K. It is worth noting that on the LVIS dataset, compared with the previous SOTA DAM, the number of parameters of WOW-Seg is reduced by nearly 9×, but it exceeds DAM by 0.7 in the Semantic Similarity. In terms of the Semantic IoU, WOW-Seg is 4.1 higher than the previous SOTA PAM, and the number of parameters is 3× lower than that of PAM. Moreover, on the RR-7K dataset, WOW-Seg continues to achieve optimal performance. Simultaneously, we observe that previous methods generally perform poorly. This further demonstrates that RR-7K presents a greater challenge than LVIS and PACO.

## 4.3 OPEN-VOCABULARY SEGMENTATION

The goal of this task is to complete the classification task for the corresponding masked area under the prompt of an open vocabulary. Typically, previous methods would input a vocabulary into the model. However, WOW-Seg is a model that does not require any text input. In the inference stage, we can use Sentence BERT (Reimers & Gurevych, 2019) to calculate the semantic similarity between the region embeddings output by WOW-Seg and the text embeddings of each category in the vocabulary, and take the category with the highest similarity as the final classification result.

Table 2 shows the performance of WOW-Seg in open vocabulary

Table 2: Recognition performance on open-vocabulary panoptic segmentation (PQ) and semantic segmentation (mIoU) upon the validation sets of Cityscapes and ADE20K. The ground truth box/mask is used for performance evaluation.

| **Method** | Cityscapes | | ADE20K-150 | |
|---|---|---|---|---|
| | PQ | mIoU | PQ | mIoU |
| CLIP-ConvNeXt-L (2021) | 22.53 | 23.06 | 36.86 | 28.74 |
| CLIP-Surgery-ViT-L (2023) | 27.24 | 21.92 | 26.55 | 21.42 |
| Kosmos-2 (2023) | 12.09 | 13.71 | 6.53 | 5.40 |
| Shikra-7B (2023) | 17.80 | 17.77 | 27.52 | 18.24 |
| GPT4RoI-7B (2024) | 34.70 | 36.73 | 36.32 | 25.82 |
| Ferret-7B (2023) | 35.57 | 38.40 | 39.46 | 31.77 |
| Osprey-7B (2024) | 50.64 | 49.78 | 41.89 | 29.63 |
| **WOW-Seg (Ours)** | **65.76** | **66.40** | **44.46** | **37.77** |

panoptic segmentation and semantic segmentation on the validation sets of Cityscapes (Cordts et al., 2016) and ADE20K (Zhou et al., 2017). The experimental results clearly demonstrate that WOW-Seg significantly outperforms existing methods in all evaluation metrics.

Our model demonstrates exceptional performance on the Cityscapes benchmark. WOW-Seg achieves a panoramic segmentation quality (PQ) of 65.76 and a semantic segmentation mIoU of 66.40. These results represent a new state-of-the-art, outperforming the previous leading method, Osprey-7B, by significant margins of +15.12 PQ and +16.62 mIoU, respectively. This superior performance is consistent across other benchmarks, with WOW-Seg also surpassing all competing

Table 3: Compare the classification accuracy of ground-truth masks (maskAcc) using different methods on multiple datasets.

| | ADE-847 | PASCAL-459 | ADE-150 | COCO Stuff | Cityscapes |
|---|---|---|---|---|---|
| ***Pretrained*** | | | | | |
| OpenAI CLIP (2021) | 32.0 | 44.8 | 51.0 | 46.2 | 46.5 |
| EVA02 CLIP (2023) | 35.2 | 44.8 | 52.7 | 45.0 | 44.9 |
| ***Finetuned*** | | | | | |
| CLIPSelf (2023) | 33.6 | 49.0 | 56.1 | 50.6 | 52.1 |
| CAT-Seg (2024) | 33.5 | 50.7 | 62.3 | 63.0 | 65.3 |
| MaskCLIP++ (2024) | 38.4 | 56.4 | 67.0 | 67.8 | 71.0 |
| **WOW-Seg (Ours)** | **59.2** | **58.7** | **68.5** | **72.0** | **73.5** |

Table 4: Performance of different variants of Cascade Attention Mask and the baseline.

| Train Setting | Cascade Attention Mask | | LVIS | | PACO | |
|---|---|---|---|---|---|---|
| | Region | Output | Sem. Sim. | Sem. IoU | Sem. Sim. | Sem. IoU |
| SM | - | - | 85.22 | 74.70 | 79.46 | 66.04 |
| MM | - | - | 88.75 | 80.10 | 86.29 | 75.49 |
| MM | ✓ | | $89.65_{(+0.90)}$ | $82.18_{(+2.08)}$ | $87.89_{(+1.60)}$ | $78.38_{(+2.89)}$ |
| MM | | ✓ | $89.56_{(+0.81)}$ | $81.94_{(+1.84)}$ | $87.36_{(+1.07)}$ | $77.42_{(+1.93)}$ |
| MM | ✓ | ✓ | $89.70_{(+0.95)}$ | $82.35_{(+2.25)}$ | $88.52_{(+2.23)}$ | $79.22_{(+3.73)}$ |

models on the ADE20K-150 dataset. Notably, these gains are achieved with a 1B parameter model, highlighting the advanced efficiency and effectiveness of our architecture compared to larger 7B models.

## 4.4 MASK REGION CLASSIFICATION

As WOW-Seg is a vocabulary-free model, we benchmark it against leading vocabulary-based VLM methods to provide a fair and comprehensive evaluation of its mask classification performance. We test using ADE20K, PASCAL (Everingham et al., 2010), COCO Stuff and Cityscapes. The results, presented in Tab. 3, demonstrate that WOW-Seg significantly outperforms all baselines across every reported metric, for both their pretrained and fine-tuned variants.

## 4.5 ABLATION STUDY

In order to evaluate the effectiveness of the core design in our work, we conducted a large number of ablation experiments.

**The Effectiveness of Cascade Attention Mask.** In the Tab. 4, SM represents training with single-mask data, and MM represents training with multi-mask data. "Region" represents decoupling between mask tokens. "Output" represents decoupling between output object names. For specific details of the variant, please refer to Fig. 10. All experiments ensure the same number of training steps. We can find that the performance of the model trained by MM is significantly higher than that of the model trained by SM. This is reasonable because the model trained by MM can learn more data in the same number of training steps. We have verified the performance of different variants of Cascade Attention Mask respectively. It can be seen that the performance of all variants is higher than that of the MM model without using Cascade Attention Mask. And the highest performance is achieved when decoupling the features of the mask area and the object name at the same time.

**The Effectiveness of Mask2Token.** In the Tab. 5, Scale denotes the multiple of the mask region image's side length relative to the mask's maximum side length in Mask2Token. Although performance is similar when scale is set to 2 and 1.5, the number of tokens required to encode a single mask is

Table 5: The impact of Region scale on the performance of Mask2Token.

| Scale | LVIS | | PACO | |
|---|---|---|---|---|
| | S. Sim. | S. IoU | S. Sim. | S. IoU |
| 1.5 | **89.91** | **82.59** | 88.18 | 78.76 |
| 2 | 89.70 | 82.35 | **88.52** | **79.22** |
| 2.5 | 89.34 | 81.61 | 88.08 | 78.32 |
| 3 | 89.13 | 81.33 | 88.34 | 78.76 |
| 3.5 | 89.58 | 81.89 | 87.79 | 77.89 |

Table 6: Ablation experiments on different mask transformation methods.

| Method | LVIS | | PACO | |
|---|---|---|---|---|
| | S. Sim. | S. IoU | S. Sim. | S. IoU |
| Fore2Token | 83.67 | 72.54 | 76.46 | 62.58 |
| Blur2Token | 82.97 | 71.28 | 78.09 | 64.35 |
| Mask2Token | **85.22** | **74.70** | **79.46** | **66.04** |

approximately $1.8\times$ greater when scale equals 2 compared to when it equals 1.5. Therefore, we have selected scale 2 for reporting the final model. To demonstrate the effectiveness of Mask2Token, we explored several of its variants during the research. Fore2Token only fills the background area with white as the mask patch. Region2Token applies Gaussian blur to the background area as the mask patch. Each mask in both cases is represented by 256 tokens. The Tab. 6 show that Mask2Token has significantly improved performance. Further details are displayed at D.1.

### 4.6 FORWARD EFFICIENCY ANALYSIS

A potential concern regarding the Mask2Token module is that encoding individual mask regions may introduce significant computational overhead. We analyse the training forward runtime and computational complexity of WOW-Seg versus PAM, as illustrated in Fig. 7. PAM employs a single-instance inference paradigm, requiring $N$ forward passes to process $N$ instances, resulting in a linear increase in computational cost. In contrast, while Mask2Token necessitates cropping and encoding each region, these operations can be efficiently batch-processed on GPUs. Crucially, LLM decoding executes only once simultaneously for all instances. Consequently, increasing the number of instances from 1 to 32 results in less than a fourfold increase in WOW-Seg's total computational cost. This confirms that Mask2Token effectively distributes the visual encoding burden, rendering WOW-Seg highly efficient.

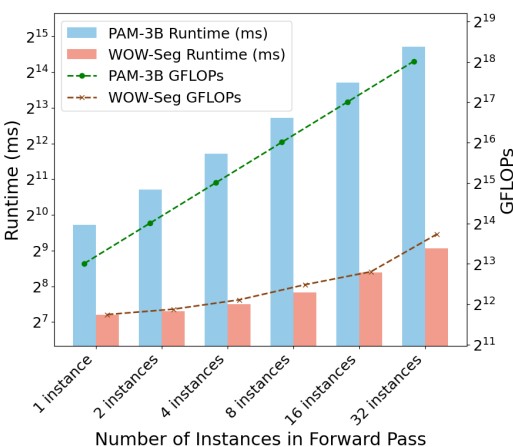

Figure 7: Comparing forward pass efficiency and GFLOPs on a 3090 GPU.

## 5 CONCLUSION

We propose WOW-Seg, a Word-free Open World Segmentation Model. WOW-Seg utilizes the Mask2Token module to represent object masks as tokens. A key advantage of this approach is that the generated tokens already lie within the VLM's pretrained embedding space, which circumvents the need for extensive retraining for feature alignment. Due to the introduction of the Cascade Attention Mask, WOW-Seg can process multiple masks in a single forward pass without being affected by the correlations between the masks. To better evaluate the model's ability in open-world segmentation tasks, we introduce the RR-7K open world test dataset. Ultimately, with only 1B parameters, WOW-Seg achieves advanced performance on multiple open world tasks.

## ACKNOWLEDGMENTS

This research was supported by the Fund of Shenzhen Science and Technology Program (QNXMB20250701090801002, JCYJ20250604184027034, JCYJ20240813114237048), the National Natural Science Foundation of China (Grant No. 62576177), Guangdong Basic and Applied Basic Research Foundation (2026A1515011435), the Fundamental Research Funds for the Central Universities 070-63253222, and the Tianjin Key Laboratory of Visual Computing and Intelligent Perception (VCIP). Computation is supported by the Supercomputing Center of Nankai University (NKSC). "Science and Technology Yongjiang 2035" key technology breakthrough plan project (2025Z053), Chinese government-guided local science and technology development fund projects (scientific and technological achievement transfer and transformation projects) (254Z0102G), National Science Fund of China under Grant No. 62361166670. This work was also supported by the Academy for Advanced Interdisciplinary Studies, Nankai University (AAIS, NKU).

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

## A    ETHICAL STATEMENT

The RR-7K dataset contributed in this work is derived from images in the SA-1B dataset, with annotations generated automatically via large model inference. The authors have conducted further manual screening and desensitization processing on the data. Our contribution is intended solely for research purposes, with the aim of advancing the study of open-world segmentation methods. The authors shall not be held responsible for any consequences arising from the practical application of the models or the dataset.

## B    REPRODUCIBILITY STATEMENT

We will provide comprehensive information to facilitate the control of randomness during training and validation, including but not limited to hardware specifications, software environment configurations, and model hyperparameters. Furthermore, we will release the complete source code alongside detailed training logs. All relevant details will be exhaustively documented in the supplementary materials.

## C    DISCLOSURE OF LLM USAGE

In the research and preparation of this manuscript, large language models (LLMs) were utilized for code implementation and language polishing. All code generated by an LLM underwent rigorous review, debugging, and comprehensive testing by the authors. The core algorithms and experimental design were conceived and executed directly by the authors. The LLM served solely as an assistive tool, with its outputs consistently under the authors' supervision and control. The authors assume full responsibility for all content presented in this work.

## D    METHOD DETAILS

### D.1    FURTHER ANALYSIS OF MASK2TOKEN

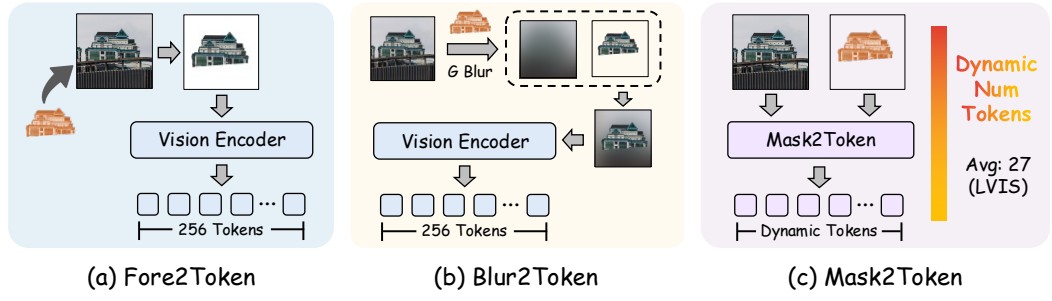

Figure 8: The process of mapping mask to tokens in different methods.

When converting the mask into features that can be understood by VLLM, we tried various methods. In Fig. 8(a), the foreground and background in the area around the mask are classified according to the mask. After filling the background with white, it is combined with the foreground. The synthesized image is then fed into the Vision Encoder. For an image, the number of tokens encoded by the Vision Encoder is fixed at 256. This became our baseline. In previous research, Gaussian blur visual cues achieved good results in open vocabulary tasks. Therefore, in Fig. 8(b), we applied Gaussian blur to the background. The standard deviation of the Gaussian blur was set to 10. Similarly, the mask processed by Blur2Token is still encoded into 256 tokens. In Fig. 8(c), we use the mask as a prior to extract visual tokens at specific positions. In this way, the number of mask tokens is allowed to change dynamically. Taking the LVIS dataset as an example, each mask is represented by an average of 27 tokens. Compared with the previous two methods, Mask2Token not only greatly reduces the number of tokens but also eliminates interference information. Fig. 9 shows the loss decline trends of these three different methods. It can be seen that Mask2Token converges faster and has the lowest loss.

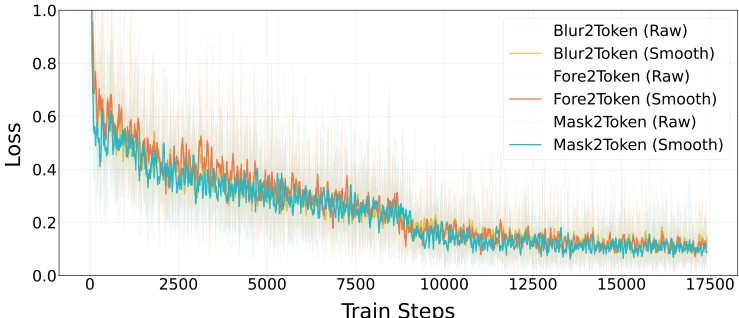

Figure 9: Comparison of the loss decline trends of Mask2Token, Fore2Token, and Blur2Token. For intuitive display, we truncated the y-axis at 1.0. In the figure, the light colored lines represent the actual loss values, and the dark colored lines are the smoothed loss decline curves.

## D.2 ROBUSTNESS TO VISUAL PROMPT QUALITY

Whilst our primary experiments use GT masks to test recognition performance, real-world applications typically rely on masks or heuristic geometries from generators. To assess WOW-Seg's robustness to imperfect prompts, we conduct comprehensive stress testing with diverse mask sources, ranging from high-quality generators to coarse geometric approximations. As shown in Table 7, the results demonstrate remarkable robustness. When using masks generated by SAM2-L, WOW-Seg maintains nearly identical performance to the ground truth baseline. More remarkably, even when provided with extremely coarse inputs such as rotated bounding boxes, the model retains robust recognition capabilities, achieving a semantic similarity of 87.3 and semantic IoU of 78.6 on

Table 7: Robustness analysis on different visual prompt sources. RBB denotes the use of a rotating bounding box as a prompt mask. Ellipse denotes the use of a bounding ellipse as a prompt mask.

| Mask Source | LVIS | | PACO | |
|---|---|---|---|---|
| | S. Sim. | S. IoU | S. Sim. | S. IoU |
| ***Geometric Heuristics*** | | | | |
| GT Mask | 89.7 | 82.4 | 88.5 | 79.2 |
| RBB | 87.3 | 78.6 | 85.5 | 74.1 |
| Ellipse | 86.9 | 78.0 | 84.8 | 73.4 |
| ***Mask Generator*** | | | | |
| SAM (Huge) | 88.8 | 81.0 | 86.6 | 75.7 |
| HQ-SAM (Huge) | 88.9 | 81.1 | 86.2 | 75.3 |
| SAM2 (Large) | 89.0 | 81.2 | 86.8 | 76.3 |

LVIS. This confirms WOW-Seg's ability to effectively utilise visual prompts for localisation without being overly sensitive to mask precision.

## D.3 CASCADE ATTENTION MASK

In the Region Attention Mask variant in Fig. 10 (a),only the masks are set to be independent of each other. A normal Causal Attention Mask is used during output. In the Output Attention Mask variant in Fig. 10 (b), only the object names are set to be independent of each other. A normal Causal Attention Mask is used for the multi-mask features during forward propagation.

The Cascade Attention Mask has a wide range of application scenarios. For example, in the visual grounding task, it ensures that the output coordinates of multiple targets do not interfere with each other. We hope that more researchers will continue to explore. Therefore, we provide the pseudocode of the Cascade Attention Mask, as shown in Algorithm 1.

## D.4 THE IMPACT OF VISUAL LANGUAGE MODEL FOUNDATIONS

To investigate the influence of pre-trained foundation models on WOW-Seg, we evaluated WOW-Seg across different iterations of the InternVL series (InternVL2, InternVL2.5, and InternVL3). As shown in Table 8, WOW-Seg demonstrates exceptional scalability and architectural superiority. Specifically, when equipped with the earlier InternVL2-1B (released July 2024), our model achieved a semantic similarity of 88.8 and semantic IoU of 80.8 on LVIS. These results still significantly outperform PAM's 87.4 and 76.5, confirming that our framework's effectiveness is intrinsic rather than

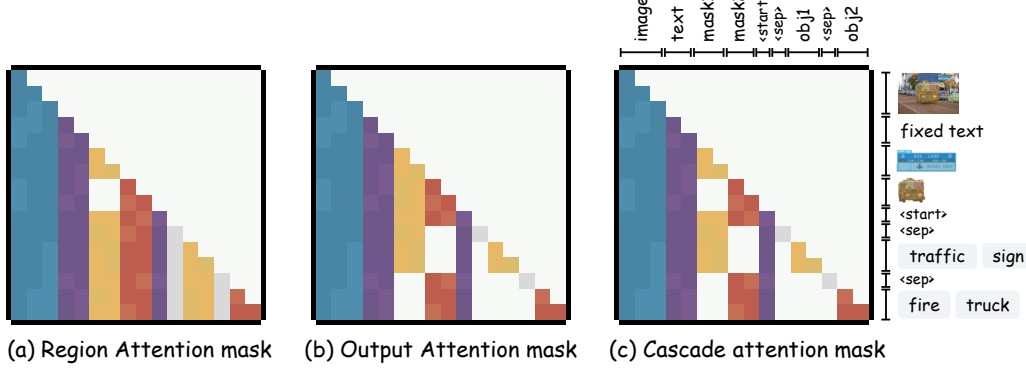

Figure 10: Cascade Attention Mask and its variants.

---

**Algorithm 1:** Cascade Attention Mask Construction

---

**Input:** Input tokens $S$, config $\mathcal{C}$
**Output:** Additive attention mask $M \in \mathbb{R}^{n \times n}$
Initialize binary causal mask $M \leftarrow \text{tril}(\mathbf{1}^{n \times n})$, where $n = |S|$ ;
Parse $S$ into sets: $\mathcal{I}_{\text{mask}}$ (mask tokens), $\mathcal{I}_{\text{out}}$ (ordered output chunks);
**for** *each mask segment* $s \in \mathcal{I}_{mask}$ **do**
    $M[s, \mathcal{I}_{\text{mask}} \setminus s] \leftarrow 0$ ;              # Mask inter-segment attention
**end for**
**for** *each output chunk* $c \in \mathcal{I}_{out}$ **do**
    $M[c, \bigcup_{k<c} k] \leftarrow 0$ ;              # Mask previous output chunks
    **if** *not* $\mathcal{C}.output\_sees\_mask$ **then** $M[c, \mathcal{I}_{\text{mask}}] \leftarrow 0$;
**end for**
Set $M[i, :] \leftarrow 0$ for $i \in$ separator/padding indices;
**return** $(1 - M) \cdot (-\infty)$ ;              # Convert to additive mask

---

merely a consequence of a more advanced backbone. Furthermore, performance steadily improves as the backbone is upgraded to InternVL2.5 and InternVL3, demonstrating that the WOW-Seg architecture effectively unlocks the potential of increasingly powerful foundational models.

Table 8: The Impact of Different Periods' Base Models on WOW-Seg. Owing to the unique structure of PAM, it is incapable of performing multi-instance inference during a single forward pass.

| Method | Base Model | Multi-Instance | LVIS | | PACO | |
|---|---|---|---|---|---|---|
| | | | S. Sim. | S. IoU | S. Sim. | S. IoU |
| PAM-1.5B | SAM2+Qwen2.5 | × | 87.4 | 76.5 | 85.1 | 73.5 |
| WOW-Seg | InternVL2 | ✓ | 88.8 | 80.8 | 85.4 | 74.0 |
| | InternVL2.5 | ✓ | 89.5 | 82.0 | 87.8 | 77.8 |
| | InternVL3 | ✓ | **89.7** | **82.4** | **88.5** | **79.2** |

# E    RR-7K DATASET

## E.1    COARSE CATEGORY LIST

To more intuitively reflect the richness of RR-7K, we have roughly classified the categories of RR-7K at a coarse grained level and presented the relevant information of the coarse grained categories, as shown in Tab. 9.

Table 9: A preliminary coarse grained division of RR-7K

| Coarse category | Fine-grained category |
|---|---|
| **Animals** | |
| Mammals | lion, bull, tiger, buffalo, elephant, cow, kangaroo, sheep, horse, dog, elk, boxer, antelope, cattle, lioness, orangutan, pony, lemur, boar, hippopotamus, yak, polar bear, chimpanzee, gazelle, llama, hedgehog, wildebeest, ... |
| Birds | bird, eagle, duck, rooster, swan, parrot, pigeon, vulture, barnacle goose, toucan, pelican, hawk, peacock, egret, ibis, cardinal, ostrich, crow, woodpecker, gull, lorikeet, seagull, cormorant, dove, ... |
| Reptiles | turtle, snake, iguana, tortoise, serpent, crocodile, gecko, komodo dragon, triceratops |
| Fish | fish, clownfish, sardine, carp, goldfish, catfish, swordfish, guppy, pufferfish |
| Insects | butterfly, ladybug, insect, moth, grasshopper, caterpillar, mosquito |
| Invertebrates | jellyfish, octopus, scallop, slug, snail, sea anemone, clam, worm, brain coral, mussel, sea urchin, coral polyp, horseshoe crab, nudibranch, prawn, ... |
| Humans | audience, audience member, child, girl, soldier, boy, firefighter, monk, adult, baby, pedestrian, tennis player, fisherman, priest, wrestler, warrior, bride, referee, gymnast, tourist, groom, spectator, teacher, knight,, ... |
| **Plants** | |
| Flowers | flower, sunflower, tulip, lotus, anthurium, bougainvillea, orchid, poppy, hibiscus, lily, peony, cherry blossom, iris, daffodil, banana flower, carnation, zucchini flower, daisy, petunia, rhododendron, rose, chrysanthemum, ... |
| Ornamental Plants | flower bed, plant, garden, lavender field, flower seed, laurel, seedlings, hydrangea, ... |
| Climbing Plants | grapevine, wisteria, vine |
| Medicinal Plants | yucca plant, aloe, cannabis plant |
| Spice Plants | pepper, chili, ginger, turmeric, cinnamon sticks, galangal, dried chili peppers, dried tobacco leaves |
| Edible Plants | seed pod, agave, seedpod, strawberry leaf, seed, shallots |
| Nut Plants | almond, chestnut |
| Herbal Plants | herbal medicine |
| Trees | palm, pine, cherry blossom tree, cypress, joshua tree, banana tree, coconut tree, mangrove, ... |
| Shrubs | bush, cactus, shrub, shrub hedge, hedge, coleus, twigs, ... |
| Herbs | grass, garlic, succulent, sedum, pitcher plant, elephant ear, dandelion, cilantro, basil, cotton, lavender, moss, ... |
| Aquatic Plants | water lily, aquatic plants, aquatic vegetation, aquatic plant, water lettuce |
| Bamboo | bamboo, bamboo stick, bamboo leaf, bamboo pole |
| Fruit Plants | olive, litchi, coffee cherries, prickly pear, lotus pod |
| Vegetable Plants | onion, cucumber, strawberry, bottle gourd, cranberry, luffa, arugula, scallion, collard greens, potato |
| Agricultural Plants | corn, wheat, buckwheat, rice bundle, ... |

| Coarse category | Fine-grained category |
|---|---|
| Man-made Objects | |
| Furniture & Home Decor | basket, clock, door, chair, pillow, box, carpet, shelf, vase, bench, curtain, ... |
| Graphics & Design | frieze, cube, text, crayon, stickers, canvas, square panel, mercedes benz logo, header, motif, plaid pattern, ... |
| Tools & Equipment | fan, gauge, handle, padlock, net, nail, control panel, chain, machine, knob, motor, ... |
| Vehicles & Transportation | car, truck, wheel, bus, bicycle, train, van, ferry, cruise ship, jeep, cable car, ... |
| Electronics | light, air conditioner, lamp, camera, phone, robot, wire, display screen, horn, ... |
| Buildings | building, chimney, dome, window, floor, spire, pillar, roof, skyscraper, monument, church, window pane, temple, staircase, shed, ... |
| Advertising & Marketing | banner, button, ad board, poster, label, balloon, billboard, advertisement board, menu, neon sign, card, display, mosaic, ... |
| Health & Hygiene | face mask, lotion, napkin, cosmetic product, knee pad, face shield, sponge, dispenser, first aid kit, surgical drape, soap, stethoscope, surgical gown, ... |
| Signs & Symbols | sign, flag, plaque, traffic light, number, traffic sign, tactile paving, inscription, symbol, barcode, banner sign, icon, ... |
| Personal Items | shoe, umbrella, helmet, pants, shirt, shorts, jacket, jeans, glove, trousers, bag, scarf, cap, handbag, ... |
| Food & Beverages | |
| Fruits | apple, fruit, pineapple, grape, strawberry, watermelon, banana, cherry, kiwi, orange, ... |
| Nuts & Seeds | nuts, seeds, hazelnuts, kernel, walnuts, coffee beans, pumpkin seeds, lotus seeds, sunflower seeds, pistachios, ... |
| Condiments | chili sauce, condiment bottle, soy sauce, chili sauce bottle, ketchup, mayonnaise, condiment, sauce packet, topping, salsa, thai chili paste, ... |
| Baked Goods | bread, pancake, biscuit, gingerbread house, bun, crepe, gingerbread, panettone, cinnamon roll, quiche, ... |
| Ready-to-Eat Foods | food, tea bag, soup, tempura, doner kebab, dumpling, burrito, stew, food can, taco, natto, ... |
| Vegetables | onions, vegetables, dried tomato, dried mushrooms, bok choy, cherry tomatoes, carrots, okra, spinach, sweet corn, cabbage, squash, legumes, dried vegetables, radicchio, ... |
| Drinks | coconut milk, wine, whiskey, milkshake, cola, beverages, root beer, bubble tea, coca cola, ... |
| Snacks | candy, cookie, snack packet, cracker bread, potato chip, chewing gum, gummy bear, pretzels, snack chip, fritter, tortilla chip, ... |
| Staple Foods | tofu, bagel, potato bread, spaghetti, rice bag, patty, flatbread, matzo, muesli, corn kernel, lentils, mashed potatoes, ... |
| Desserts | cake, ice cream, fried dough, strawberry jam, choco pie, marshmallow, tea cake, waffle cone, heart shaped cookie, sweet, tiramisu, ... |
| Meat | sausage, meat, ham, skewered meat, dried meat, burger, salami, meatball, ribs, raw meat, pork knuckle, pork belly, shredded meat, pepperoni, ... |
| Seafood | dried seafood, seaweed snack, dried fish, oyster, crab leg, dried seaweed, sashimi, tuna can, unagi, mussels, salmon fillet, ... |
| Dairy Products | cheese, dairy product, dairy, coffee creamer, milo product, milk powder, yoghurt |

| Coarse category | Fine-grained category |
|---|---|
| **Arts & Entertainment** | |
| Paintings | mural, coat of arms, fresco, graffiti, landscape painting, religious artwork, portrait, frame painting, altarpiece, religious mural, islamic calligraphy, framed panel, ... |
| Entertainment Venues | playground, christmas market stall, theater, cinema, casino, amusement ride, bullring, backdrop, ball-pit, arcade, ... |
| Handicrafts | ornamental carving, figurine, tarot card, snow sculpture, dreamcatcher, religious ornament, crochet, matryoshka doll, maneki neko, chinese knot, batik, souvenir, ... |
| Festivals & Celebrations | tinsel, maypole, streamers, festival float, papel picado, snowflake ornament, chinese new year decoration, offering, festival costume, confetti, christmas |
| Sculptures | statue, carving, sculpture, statue base, mermaid statue, cherub, dog statue, crosier, sphinx statue, buddha statues, clay sculpture, bird statue, relief sculpture, ... |
| Movies | film, anime characters, clapper, superhero |
| Music | trombone, harp, drum, bass guitar, double bass, bagpipes, banjo, wind instrument, lyre, marimba, musical note, brass instrument, ... |
| Books | book, stamp, word roll, writing, stacked books, book page, mickey mouse, atlas, minnie mouse, book edge, ... |
| Performing Arts | dancer, marching band member, puppet, figures, juggling club, ballerina, tango, artist, stage, ... |
| Video Games | dragon puppet, character figure, video game cover, game, spongebob squarepants, ... |
| Visual Arts | ellipse, mandala, abstract art, spiral artwork, polygon, figurines, abstract painting, cutout figure, arabesque, crystal ball, sketch, palette, ... |
| Culture & History | hieroglyphics, ushnisha, deity, prayer card, ganesha, cartouche, aztec calendar, religious figure, virgin mary, prayer tablet |
| **Environment** | |
| Geography & Landmarks | hill, path, mountain, ocean, obelisk, stadium, gravestone, bridge, stupa, ... |
| Materials | wood, fabric, cloth, plastic, concrete block, metal, wooden post, wooden surface, ... |
| Weather & Climate | sky, snow, smoke, sun, snowbank cloud, crescent, sunburst, shade, cloud, ... |
| Reflections & Optical Phenomena | reflection, shadow, water spray, lightning bolt, smoke trail, flash, bubble, laser beam, skyline reflection, water surface, light trails, ... |
| Natural Landscapes | water, star, sand, pool, rock formation, ground, ocean water, stalagmite, ... |

### E.2 THE SEMANTIC SPACE OF RR-7K

To demonstrate that the 7,662 categories in RR-7K represent a broad semantic space rather than localized lexical inflation, we conduct a semantic space comparison between RR-7K and ImageNet-21K (Deng et al., 2009), as shown in Fig. 11.

We use BERT (Reimers & Gurevych, 2019) to extract text embeddings for all category names in RR-7K and ImageNet-21K, then reduce the dimensionality to a two-dimensional plane via the t-SNE algorithm for visualisation.

As can be seen from Fig. 11, RR-7K exhibits semantic breadth comparable to ImageNet-21K alongside a similar distributional topology. This demonstrates that the 7,000 categories within RR-7K do not constitute a mere repetition of a small number of concepts, but rather construct a diverse semantic space.

### E.3 HIERARCHICAL DISTRIBUTION OF RR-7K

We further present bar charts illustrating the sample distributions for the primary and secondary categories, as shown in Fig. 12.

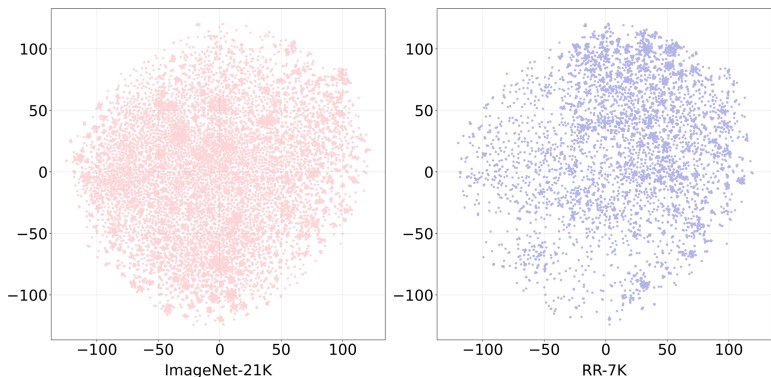

Figure 11: Visualisation of category distributions on ImageNet-21K and RR-7K using t-SNE.

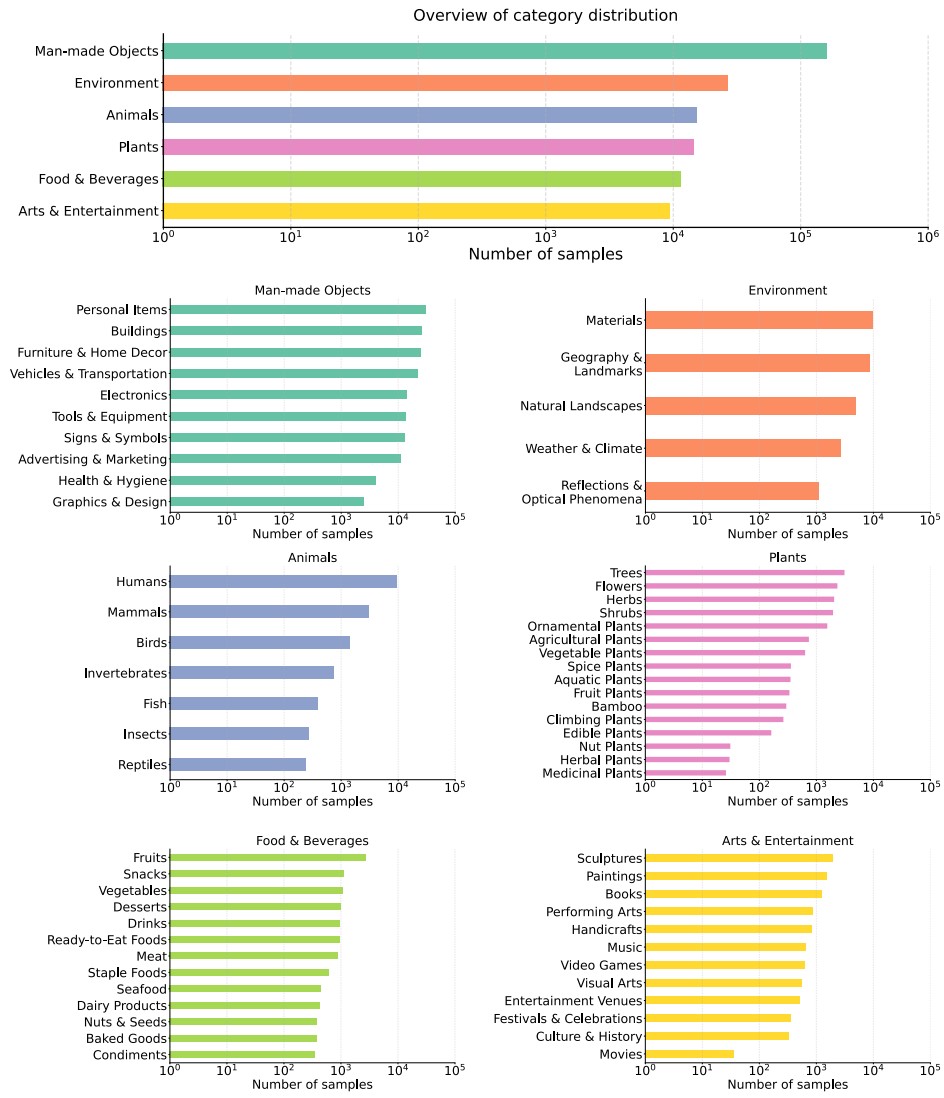

Figure 12: Hierarchical distribution visualisation. The top panel displays the distribution of primary categories. The remaining six sub-panels each show the distribution of secondary categories.

Statistical analysis indicates that the dataset exhibits extensive coverage across all major semantic domains while maintaining reasonable internal diversity. This demonstrates that RR-7K possesses exceptional richness in both breadth and depth.

### E.4 VISUALIZATION OF RR-7K

In Fig. 13, we present the visualization of part of the RR-7K data.

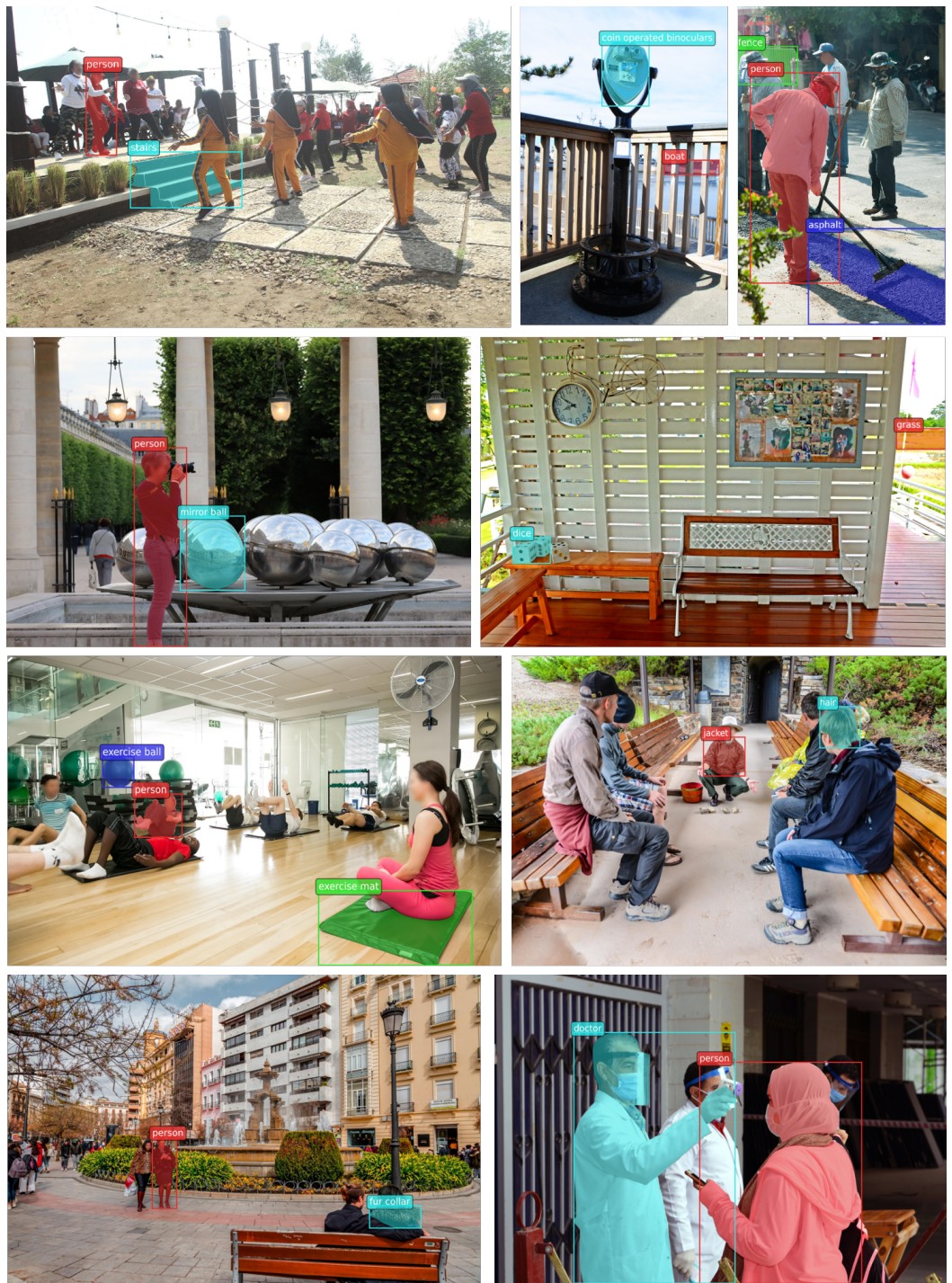

Figure 13: Visualization of some rare category objects in RR-7K.

## F    MORE PERFORMANCE OF WOW-SEG

In Fig. 14, we visualise the regional recognition performance of WOW-Seg on the ADE20K dataset. It is noteworthy that WOW-Seg is not trained on the ADE20K dataset.

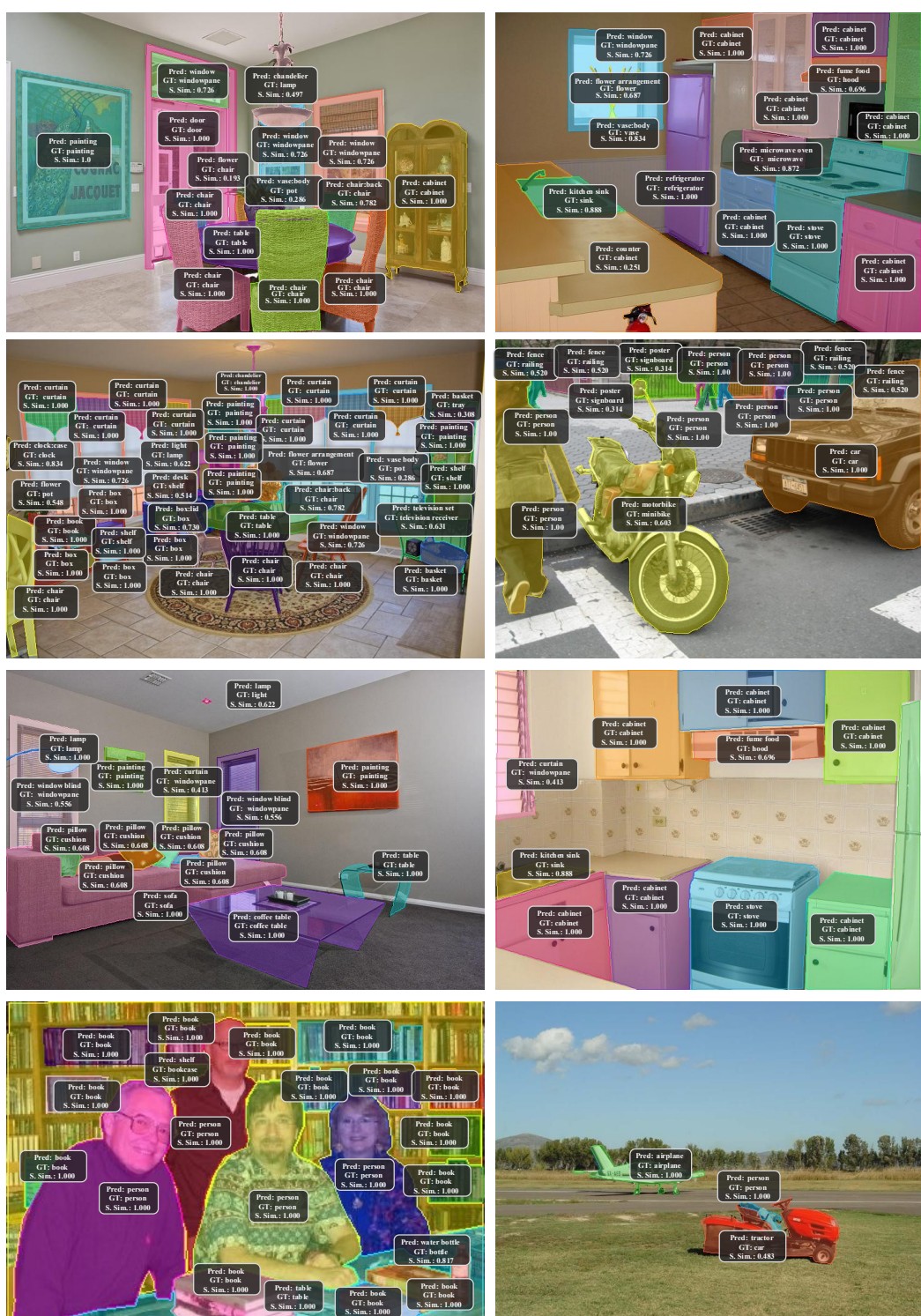

Figure 14: Visualisation results of WOW-Seg on the ADE-20K dataset.

In Fig. 15 and Fig. 16, we visualized the performance of WOW-Seg-1B and the previous SOTA method PAM-3B. It can be seen that WOW-Seg has good recognition performance on small targets, super large targets, and rare targets.

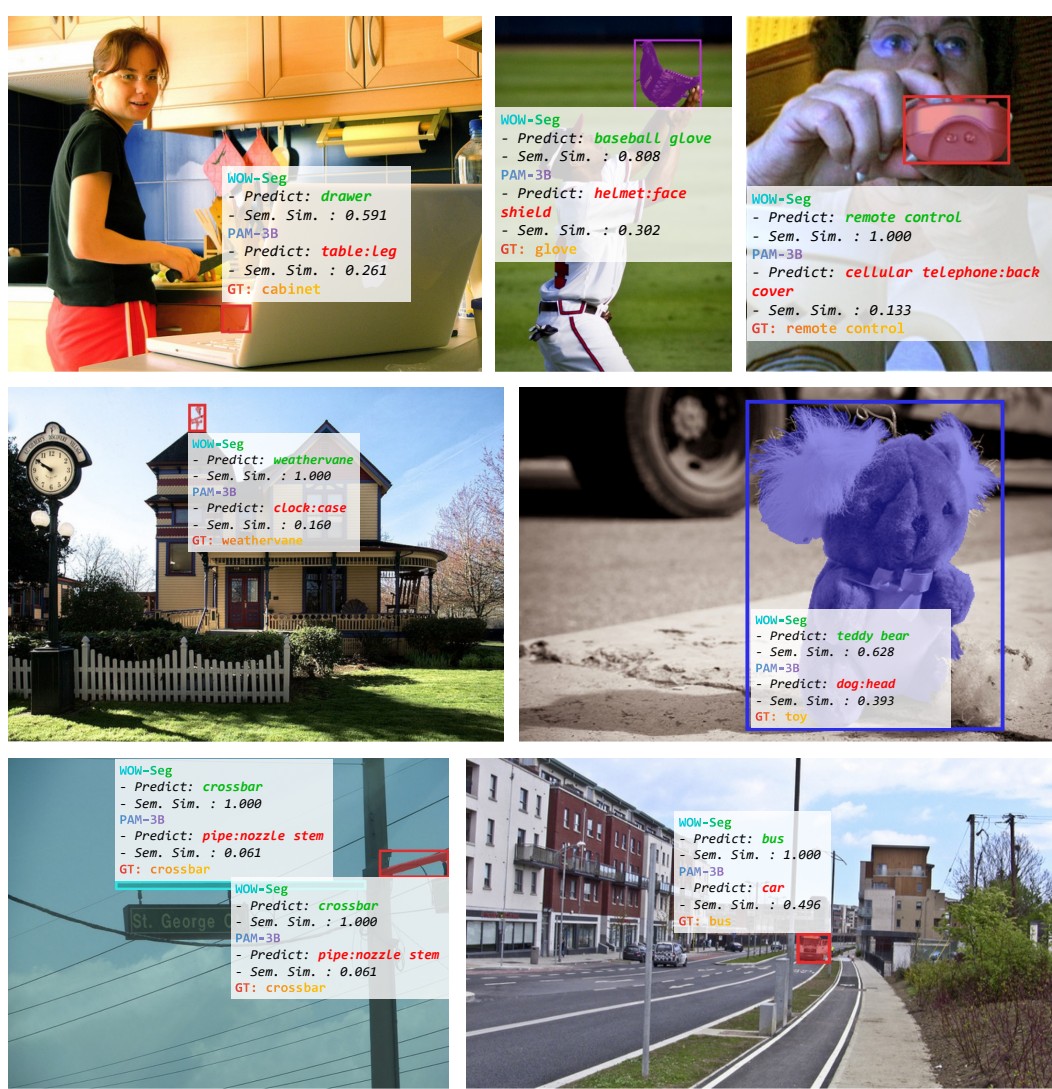

Figure 15: Visualization of the performance of WOW-Seg-1B and PAM-3B.

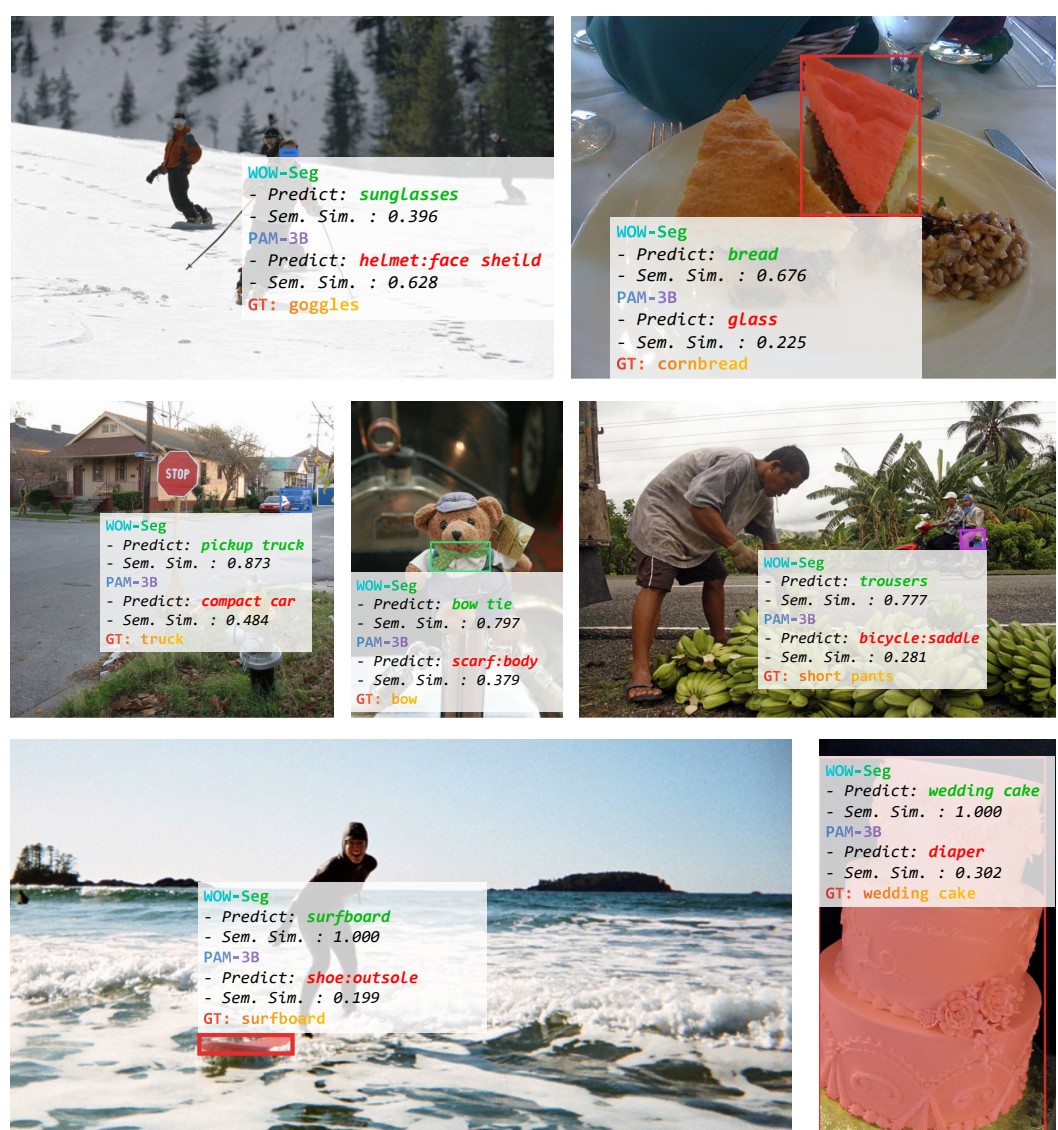

Figure 16: Visualization of the performance of WOW-Seg-1B and PAM-3B.

