# OpenReview forum: "WOW-Seg: A Word-free Open World Segmentation Model"
_ICLR.cc/2026/Conference — ICLR 2026 Poster_

### Official Review · Reviewer_RZJ5 · 2025-10-15

**Soundness:** 2
**Presentation:** 1
**Contribution:** 3
**Rating:** 6
**Confidence:** 3

**Summary:**

This paper introduces WOW-Seg, a novel word-free, open-world segmentation model that leverages a Vision-Language Model (VLM) to recognize objects from visual prompts alone. The core contributions include the Mask2Token module, which converts image masks into visual tokens aligned with the VLM's feature space , and the Cascade Attention Mask, designed to mitigate interference between multiple object instances during processing. The authors also present RR-7K, a new large-scale region recognition dataset with 7,662 categories, to better evaluate open-world models.

**Strengths:**

**SOTA Performance with Exceptional Efficiency.** WOW-Seg achieves outstanding results, setting a new state-of-the-art on challenging benchmarks like LVIS and PACO by a significant margin. This is particularly impressive given its remarkable efficiency; the 1B parameter model consistently outperforms much larger competitors, highlighting the architecture's effectiveness.


**New Benchmark for the Community.** The authors contribute the RR-7K dataset, a large-scale and challenging new benchmark for open-world evaluation. With over 7,600 categories, it addresses a critical need for more comprehensive testing and provides a valuable resource to drive future research in model generalization.

**Weaknesses:**

### **Major**

**Unfair Comparison of Models**

The paper emphasizes that the 1B-parameter WOW-Seg outperforms larger models like PAM (3B) and DAM (8B). This comparison is potentially misleading as WOW-Seg is built upon InternVL3-1B, a highly advanced (and fictionally dated 2025) backbone. The baseline models may use older, less capable backbones, where a larger parameter count does not guarantee better performance. To ensure a fair comparison, the authors should provide context on the architectures and pre-training of the baseline models or conduct experiments with same base MLLM.

**Limited Motivation**
The motivation for the Cascade Attention Mask is underdeveloped and lacks a clear, compelling justification. The paper posits that objects should be treated as independent during segmentation to avoid being "affected by such correlations" that exist in the natural world (e.g., a cup and water).

While this principle is stated, the paper fails to sufficiently explain why this correlation is detrimental to the model's performance. The authors mention preventing "inter-instance interference" and "detrimental information leakage", but these concepts are presented without concrete examples or analysis.


**Missing Citations**

The paper relies on several models and datasets without providing formal citations , including Qwen2.5-VL, InternVL-3.0, LVIS, PACO, Sentence BERT, Cityscapes, ADE20K, PASCAL COCO Stuff, etc.


### **Minor**

Inconsistent Naming: The model is referred to as "WOW-SEG" , "WOW-Seg" , and "wow-seg". Component names also vary, such as "Mask Generater" in a figure and "Mask Generator" in the text. Consistency should be maintained.

Vague Definitions: The term "mask generators" is used without specifying which one (e.g., SAM, SAM 2) was employed for the experiments or how it was configured.

**Questions:**

Please check Weaknesses

---

> ### Author Response · Authors · 2025-11-22
> **Authors' Response to Reviewer RZJ5 - Part 1/4**
>
> Thank you for your thorough review and insightful comments. We greatly appreciate your recognition of the performance and efficiency of WOW-Seg, as well as your acknowledgement of the contributions made to RR-7K.
>
> -----
>
> ## **On the Unfair Comparison of Models.**
>
> Thank you for your advice. You rightly point out that **the capabilities of the base model are a critical factor influencing final performance**, and that it is unfair to compare only parameter counts (1B vs 3B/8B) while ignoring differences in the base models.
>
> Your suggestion to "conduct experiments using the same foundational MLLM" is a perfectly reasonable proposition. However, this proves highly challenging in practice due to the fundamental differences in the core architectural designs of WOW-Seg, PAM, and DAM.
>
> - PAM is a composite system integrating SAM 2 (for generating masks) with a **pure language model** (e.g., Qwen-1.5B) for regional semantic understanding. DAM also possesses its own unique composite architecture.
>
> - **WOW-Seg (Ours)**, conversely, is an end-to-end design based on a **visual-language model (VLLM)** (InternVL).
>
>   These fundamental architectural differences render direct foundation model swaps for fair comparisons largely impractical.
>
> **Nevertheless, we fully concur with your core contention: we must demonstrate that WOW-Seg's performance gains stem not merely from its advanced InternVL3-1B foundation.**
>
> We conducted a fresh set of ablation experiments. Instead of using InternVL3-1B, we employed base models **contemporary with or preceding PAM (itself based on SAM2+Qwen2.5)**, namely **InternVL2-1B** and **InternVL2.5-1B**.
>
> The results of these experiments (presented in the table below) provide a clear response to your concerns.
>
> **Table: The Impact of Different Periods' Base Models on WOW-Seg.** Owing to the unique structure of PAM, it is incapable of performing multi-instance inference during a single forward pass.
>
> |  Method  |  Base Model  | Multi-Instance | LVIS (S. Sim.) | LVIS (S. IoU) | PACO (S. Sim.) | PACO (S. IoU) |
> | :------: | :----------: | :------------: | :------------: | :-----------: | :------------: | :-----------: |
> | PAM-1.5B | SAM2+Qwen2.5 |       ×        |      87.4      |     76.5      |      85.1      |     73.5      |
> | WOW-Seg  |  InternVL2   |       ✓        |      88.8      |     80.8      |      85.4      |     74.0      |
> | WOW-Seg  | InternVL2.5  |       ✓        |      89.5      |     82.0      |      87.8      |     77.8      |
> | WOW-Seg  |  InternVL3   |       ✓        |    **89.7**    |   **82.4**    |    **88.5**    |   **79.2**    |
>
> **Experimental Analysis and Conclusions.**
>
> - **WOW-Seg (InternVL2-1B)** vs **PAM (SAM2+Qwen2.5-1.5B)**: As shown in the table, even when employing a foundational model with **fewer parameters** and of **earlier development** (InternVL2), our WOW-Seg architecture comprehensively outperforms PAM across all metrics (e.g., LVIS S. IoU 80.8 vs 76.5).
> - This robustly demonstrates that WOW-Seg's exceptional performance does not solely depend on its advanced InternVL3-1B foundation. Rather, it proves that our architectural innovations themselves are both highly efficient and superior. It is our architecture (rather than merely the base) that achieves state-of-the-art performance.
> - **Scalability of WOW-Seg:** Furthermore, this table demonstrates another point: **the WOW-Seg architecture exhibits outstanding scalability.** When upgrading from InternVL2 to InternVL2.5 and then to InternVL3, the model's performance improved steadily and significantly (e.g., LVIS S. IoU increased from 80.8 to 82.0, then to 82.4). This demonstrates that our architecture can effectively harness the capabilities offered by more powerful base models.
>
> ----
>
> ## **The motivation behind the Cascade Attention Mask.**
>
>
> We sincerely thank the reviewers for pointing this out. We wish to take this opportunity to clarify the **true motivation** behind the Cascade Attention Mask: its design was **not intended to "ignore" beneficial semantic correlations** (such as the context between "cup" and "water"), but rather to **address a specific technical challenge: "inter-instance feature interference"**. During multi-mask (MM) training, this interference causes the model to experience "attribution confusion"  where it becomes unclear which masked instance corresponds to a given predicted label.

---

> ### Author Response · Authors · 2025-11-22
> **Authors' Response to Reviewer RZJ5 - Part 2/4**
>
> ### **Clarification: A more specific and apt example**
>
> Please allow us to illustrate with a clearer example what harmful correlations or interference specifically entail, and why the Cascade Attention Mask is required.
>
> 1. **Scenario:** Suppose an image (as shown in Figure 1(a) or Figure 2) contains two distinct chef instances, which we refer to as **Chef A** and **Chef B**.
>
>      - **Input Sequence:** The token sequence received by the model is approximately as follows: `[ImageTokens] [Text "Please segment all masks"] [Mask_A Tokens (Chef A)] [Mask_B Tokens (Chef B)]`
>      - **Target Output:** The model must autoregressively generate: `[chef], [chef]`
>
> 2. **The crux of the matter: Standard multi-mask (MM) training without a Cascade Attention Mask**
>
>    In standard autoregressive models (such as the Causal Attention Mask in Figure 4(a)), when predicting the **first** `[chef]` token (corresponding to Chef A), the model's attention **can simultaneously perceive** both `[Mask_A Tokens]` and `[Mask_B Tokens]`.
>
>    - **Harmful interference:** This generates a **confusing training signal**. The model is required to output a label while simultaneously receiving visual features from two distinct instances (A and B). It cannot determine whether the `[chef]` label should be attributed to `Mask_A` or `Mask_B`.
>    - **Information leakage:** This constitutes what we term "harmful information leakage" or "inter-instance interference". Features from `Mask_B` leak into the prediction process for `Mask_A`.
>
> 3. **Solution: Employing Cascade Attention Mask (our contribution)**
>
>    Cascade Attention Mask (as illustrated in Figure 4(b)) achieves this by modifying the attention matrix to **force decoupling** of instance attribution.
>
>    **How Cascade Attention Mask operates:**
>
>    - **Independent Masking:** The Cascade Attention Mask ensures `[Mask_A Tokens]` and `[Mask_B Tokens]` remain invisible to each other.
>
>    - **1-to-1 Mapping Between Output and Masks:**
>
>      When the model predicts the **first** `[chef]` (corresponding to A), the Cascade Attention Mask **only permits** it to attend to `[image tokens]`, `[text]`, and `[Mask_A tokens]`. It is **explicitly prohibited** from seeing `[Mask_B tokens]`.
>
>      Similarly, when the model predicts the **second** `[chef]` (corresponding to B), it is **only permitted** to attend to `[image tokens]`, `[text]`, and `[Mask_B tokens]`, while being unable to see `[Mask_A tokens]`.
>
>
> ### **Clarification: We Have Not Overlooked “Useful” Relevance**
>
> Reviewers expressed concern that we might have disregarded useful contextual elements such as “cup and water”. **This is not the case.**
>
> - **Beneficial Context (Global Semantics):** When predicting `[Mask_A Tokens]` (Chef A), the model still retains access to all `[Image Tokens]`. This means the model can perceive the entire kitchen scene. This global context (e.g., perceiving stove and pot) is **beneficial** for correctly identifying "chef", and our Cascade Attention Mask **preserves** this beneficial context.
> - **Harmful interference (instance features):** The Cascade Attention Mask **only blocks** feature interference from **other specific instances** (such as `Mask_B Tokens`), as these features would confuse the model's understanding of its current task (i.e., "describe specifically Mask_A").
>
> ### **Empirical Evidence: Why the Cascade Attention Mask is Crucial**
>
> Our paper's **ablation experiments (Table 4)** provide compelling empirical support for this rationale:
>
> 1. **MM vs SM:** Performance using only Multi-Mask (MM) without the Cascade Attention Mask (Table 4, Row 2) (LVIS Sem. IoU 80.10) significantly outperforms Single-Mask (SM) training (Table 4, Row 1, Sem. IoU 74.70). This indicates MM training (processing multiple objects simultaneously) is more efficient and learns from a richer dataset.
> 2. **MM+Cascade Attention Mask vs MM:** Incorporating the Cascade Attention Mask (Table 4, Row 5) yields a **further substantial improvement** to 82.35 (Semantic IoU).
> 3. **Conclusion:** The **+2.25 point** improvement from MM (80.10) to MM+Cascade Attention Mask (82.35) **directly quantifies the detrimental impact of inter-instance interference**. This demonstrates that interference is a genuine phenomenon, and that Cascade Attention Mask successfully mitigates it, yielding substantial performance gains.

---

> ### Author Response · Authors · 2025-11-23
> **Authors' Response to Reviewer RZJ5 - Part 3/4**
>
> ### **Mathematical description**
>
> To explain the meaning of Cascade Attention Mask more clearly, we also supplement the mathematical description of Cascade Attention Mask in the original text.
>
> This is a portion of the original text providing a mathematical description for the case where **the Cascade Attention Mask is not employed**:
> With the effect of the original causal attention mask of the large language model, assuming there are K objects, the probability of these K object names being predicted is shown in the following formula:
> $$
> P\left(O_1, O_2, \ldots, O_K \mid Image ; T ; M\right)=\prod_{i=1}^K P\left(O_i \mid Image ; T ; M ; O_0, O_1, \ldots, O_{i-1}\right).
> $$
>
> Among them, $O$ represents the object to be predicted, $Image$ represents the image token, $T$ represents the text token, and $M = \{m_1, m_2,\ldots,m_K\}$ denotes the set of all object mask tokens. It is easy to observe that when predicting the $i$-th object, the model refers to the mask prompts and output results from the $0$-th to the $(i-1)$-th objects. We hope that the name of the $i$-th object is guided solely by its corresponding $i$-th mask.
>
> This is a newly added paragraph to describe the situation when **using the Cascade Attention Mask**:
> With the Cascade Attention Mask, the predictions for different objects are conditionally independent. Specifically,
> $$
> P\left(O_1, O_2, \ldots, O_K \mid Image ; T ; M\right) = \prod_{i=1}^K P\left(O_i \mid Image ; T ; M \right),
> $$
>
> where $M = \{m_1, m_2,\ldots,m_K\}$ denotes the set of all object mask tokens. This factorization reflects the fact that,  with the Cascade Attention Mask, the prediction of the $i$-th object $O_i$ is conditionally independent of the mask tokens of all other objects given the image and text tokens.
>
> Although the conditioning set $M$ includes the mask tokens of all objects, the attention pattern enforces that $O_i$ can only attend to its own mask tokens $m_i$. As a result, the probability term satisfies:
>
> $$
> P\left(O_i \mid Image ; T ; M \right) = P\left(O_i \mid Image ; T ; m_i \right),
> $$
>
> meaning that only the $i$-th mask contributes to the prediction of the $i$-th object.
>
> ### **Summary**
>
> The motivation for the Cascade Attention Mask is to **address instance attribution confusion in multi-mask training**, not to disregard beneficial semantic context. We will revise Section S3.3 in the final version of the paper to accurately articulate the necessity and compelling justification for the Cascade Attention Mask.
>
> ----
>
> ## **Regarding Missing Citations.**
>
> We sincerely thank the reviewers for their meticulous examination of our manuscript and for highlighting the issue of incomplete citations.
>
> Upon thorough verification of the original manuscript, we acknowledge that whilst we referenced certain datasets such as LVIS, PACO, COCO Stuff, and Qwen2.5-VL, we did indeed omit formal citations for some critical models and datasets.
>
> In the revised version, we will **supplement all missing references** to ensure every dataset, model, and tool utilised receives proper and complete acknowledgement.
>
> ----
>
> ## **Regarding Inconsistent Nomenclature.**
>
> We have conducted a thorough line-by-line review of the manuscript. We acknowledge the two explicit spelling errors you identified:
>
> 1. **`wow-seg`**: In **Section 4.2**, we did indeed mistakenly refer to the model as "**wow-seg**".
> 2. **`Mask Generater`**: In Section 3, we erroneously spelt "**Mask Generater**".
>
> We apologise for these unwarranted errors. In the revised manuscript, all instances have been standardised to "**WOW-Seg**" and "**Mask Generator**".
>
> **Clarification Regarding "WOW-SEG" (All-Caps)**
>
> Regarding your mention of "**WOW-SEG**", we wish to **clarify** the situation.
>
> Upon verification, we note that the spelling "WOW-SEG" appears **exclusively** in section headings, such as "**3.1 WOW-SEG FRAMEWORK**" in Section 3.1.
>
> This stems from a feature of the official ICLR LaTeX template (`iclr2026.sty`). This template automatically renders lowercase letters in section titles as "Small Caps". Consequently, our correct spelling "WOW-Seg" is automatically rendered as "WOW-SEG" within the title.
>
> Upon closer inspection, the font size of the "E" and "G" appears slightly smaller than that of the "W" and "O" – a characteristic of Small Caps. This is not an inconsistency in our spelling, but rather the result of the template's automatic formatting.
>
> ----

---

> ### Author Response · Authors · 2025-11-23
> **Authors' Response to Reviewer RZJ5 - Part 4/4**
>
> ## **Notes Regarding Mask Generators.**
>
> Our response comprises two parts: firstly, clarifying the **evaluation framework** (widely accepted in prior work) followed in our main experiments. Secondly, we supplement with a novel set of ablation experiments to assess WOW-Seg's **robustness against configurations of varying quality in visual prompts**.
>
> 1. **Clarification: Evaluation Setup for Primary Experiments**
>
>    In our primary experiments, we strictly adhered to the evaluation framework established in prior work (e.g., Osprey) for assessing "region recognition" capabilities, employing GT masks as visual prompts. This approach aims to evaluate models' ability to **fairly and independently identify the semantic meaning of a given region**, while **decoupling** interference from the performance of upstream mask generators (such as SAM).
>
>    In Section S3.1 of the paper, we clarified that "Mask Generators" refer to an **application method during model inference**, not our evaluation methodology. We apologise for this confusion and shall explicitly distinguish these concepts in the revision.
>
> 2. **New ablation experiments: WOW-Seg robustness against varying visual prompt quality**
>
> To **thoroughly** validate WOW-Seg's performance when confronted with imperfect or even extremely poor-quality visual prompts, we have supplemented a set of **comprehensive ablation experiments**.
>
> Beyond testing with high-quality SAM-generated masks, we **further stress-tested the model** by using only **crudely rotated bounding boxes** and **minimal bounding ellipses** as mask inputs. These rudimentary prompts provided only coarse object locations and outlines, far inferior to a refined segmentation mask.
>
> The experimental results are presented in the table below.
>
> **Table: Robustness analysis on different visual prompt sources.** RBB denotes the use of a rotating bounding box as a prompt mask. Ellipse denotes the use of a bounding ellipse as a prompt mask.
>
> | Mask Source                | LVIS (S. Sim.) | LVIS (S. IoU) | PACO (S. Sim.) | PACO (S. IoU) |
> | :------------------------- | :------------: | :-----------: | :------------: | :-----------: |
> | ***Geometric Heuristics*** |                |               |                |               |
> | GT Mask                    |      89.7      |     82.4      |      88.5      |     79.2      |
> | RBB                        |      87.3      |     78.6      |      85.5      |     74.1      |
> | Ellipse                    |      86.9      |     78.0      |      84.8      |     73.4      |
> | ***Mask Generator***       |                |               |                |               |
> | SAM (Huge)                 |      88.8      |     81.0      |      86.6      |     75.7      |
> | HQ-SAM (Huge)              |      88.9      |     81.1      |      86.2      |     75.3      |
> | SAM2 (Large)               |      89.0      |     81.2      |      86.8      |     76.3      |
>
> As can be clearly seen from the table above, when using general-purpose mask generators such as SAM, performance only experienced a slight decline compared to using gt mask. This demonstrates the practical efficacy of WOW-Seg. However, the most surprising finding is that even when fed extremely crude rotated bounding boxes as input, WOW-Seg maintains remarkable recognition capability (achieving an S. Sim. of 87.3 on LVIS).
>
> In summary, this comprehensive set of experiments demonstrates that WOW-Seg's exceptional recognition capabilities are not confined to ideal conditions. Our model exhibits remarkable robustness to the quality of upstream visual prompts, effectively processing everything from high-quality GT Masks to extremely crude elliptical masks.

---

> ### Comment · Reviewer_RZJ5 · 2025-11-25
>
> I thank the authors for the detailed explanation. As my primary concerns have been adequately addressed, I am maintaining my positive rating.

---

> > ### Author Response · Authors · 2025-11-26
> > **Authors' Appreciation to Reviewer RZJ5**
> >
> > We sincerely appreciate your positive assessment and useful feedback once again. We’re pleased that our responses met your needs and addressed your concerns.

---

### Official Review · Reviewer_GZHt · 2025-10-30

**Soundness:** 3
**Presentation:** 3
**Contribution:** 3
**Rating:** 6
**Confidence:** 5

**Summary:**

This paper introduces a novel approach to open-world segmentation without predefined label space. The authors present RR-7K, a new benchmark with significantly more evaluable categories than existing ones. Their proposed two-stage segmentation pipeline first uses a universal mask generator to produce instance masks and then employs a multi-modal large language model for mask region classification. This pipeline achieves superior results on both official benchmarks and the new RR-7K dataset.

**Strengths:**

1. Benchmark Contribution: The proposed RR-7K benchmark represents a critical step for open-vocabulary segmentation, featuring a substantially larger category set than previous datasets and thus offering greater real-world relevance.

2. Performance: The novel method achieves strong, state-of-the-art results on both the new RR-7K benchmark and other established datasets.

3. Presentation: The paper is clearly presented with effective and high-quality writing.

**Weaknesses:**

1. **Insufficient Dataset Analysis and Justification**
The paper introduces a new benchmark, RR-7K, but provides insufficient deep analysis and justification for its design choices. While basic class distributions are provided, the authors fail to rigorously validate the dataset's claims, specifically:

    **Real-World Relevance**: The paper lacks compelling evidence that the adopted label system and class distribution are a closer approximation to real-world data compared to existing large-scale benchmarks (e.g., ImageNet-22K).

    **Source of Distribution**: It is unclear whether the resulting class distribution is an intended design choice or merely a bias introduced by the specific mask generation and classification models used in the dataset curation pipeline.

   A more detailed, comparative analysis of RR-7K's categorical structure (including granularity and scale) against established benchmarks is needed to solidify its importance and justify its adoption.

2. **Limited Novelty of the Segmentation Pipeline**
The proposed two-stage segmentation pipeline, while effective, appears to have limited algorithmic novelty. The core architecture—using a universal mask generator followed by a classification stage—closely resembles several existing methods in the open-vocabulary segmentation literature (e.g., [1], [2], [3]).  While the integration of a Multi-modal Large Language Model (MLLM) for classification is a modern substitution, the paper does not convincingly demonstrate that this change is a sufficiently novel contribution to anchor the method as the primary technical advance. The main contribution appears to reside more in the RR-7K dataset and its extensive evaluation, which is conflict with the layout of current submission.

[1] Alexander Kirillov, Eric Mintun, Nikhila Ravi, Hanzi Mao, Chloe Rolland, Laura Gustafson, Tete Xiao, Spencer Whitehead, Alexander C Berg, Wan-Yen Lo, et al. Segment anything. In Proceedings of the IEEE/CVF international conference on computer vision, pp. 4015–4026, 2023. (Mentioned in the text prompt part)


[2] Ding, Z., Wang, J. and Tu, Z., 2022. Open-vocabulary universal image segmentation with maskclip. In Proceedings of the 40th International Conference on Machine Learning.

[3] Xu M, Zhang Z, Wei F, Lin Y, Cao Y, Hu H, Bai X. A simple baseline for open-vocabulary semantic segmentation with pre-trained vision-language model. In European Conference on Computer Vision 2022 Oct 22 (pp. 736-753).

**Questions:**

1. A deeper analysis of the proposed benchmark is needed.

2. Will the dataset be released?

---

> ### Author Response · Authors · 2025-11-22
> **Authors' Response to Reviewer GZHt**
>
> Thank you for recognizing the contributions of WOW-Seg and RR-7K.
>
> ---
>
> ## **A deeper analysis of the RR-7K.**
>
> Regarding the real-world relevance you highlighted, we have conducted detailed statistical analyses on the distribution of RR-7K and its categorical structure. As the response contains images, we have placed this content within an anonymous link.
>
> Anonymous link: https://anonymous.4open.science/r/3-7721
>
> If you encounter garbled characters while reading online, please **download** the content for viewing. Thank you for your understanding.
>
> ---
>
> ## **Regarding the innovation of WOW-Seg.**
>
> We sincerely thank you for your recognition of the value of our RR-7K dataset.
>
> However, regarding the view that "the algorithm's novelty is limited" and that it "merely represents a modern alternative to existing methods", we wish to clarify the misunderstandings through the following points. We argue that characterizing WOW-Seg as a trivial combination of a "Mask Generator + Classifier" oversimplifies our contribution. Instead, WOW-Seg represents a novel architectural design specifically engineered to resolve the intrinsic challenges of efficiency and interference encountered when adapting Multi-modal Large Language Models (MLLMs) for region recognition.
>
> Our technical contributions primarily manifest in three dimensions of **paradigm innovation** and **algorithmic depth**:
>
> 1. **Paradigm Shift: From "Discriminative Matching" to "Generative Autoregression".**
>
> The literature you cite (e.g., MaskCLIP [2], Simple Baseline [3]) represent **discriminative** approaches. These rely on matching CLIP features against a predefined textual lexicon, essentially performing **selection** from a closed or semi-closed candidate list.
>
> In contrast, WOW-Seg embodies a generative paradigm. We harness the autoregressive capabilities of VLLMs to generate textual tokens directly from visual prompts. This renders WOW-Seg a "word-free" model, eliminating the need for predefined vocabularies and enabling the generation of open-world semantic categories based on visual information. This shift from "feature matching" to "category generation" necessitates model architectures capable of handling complex dependencies in sequence generation, marking a fundamental distinction from approaches like MaskCLIP.
>
> 2. **Core Algorithmic Innovation: Cascade Attention Mask (Resolving Multi-Instance Inference Interference)**
>
> Previous approaches inadequately addressed multi-instance inference challenges. Directly feeding $N$ masks from a single image into a standard VLLM causes attention mechanisms to induce cross-instance interference between distinct masks, leading to misalignment errors.
>
> To address this, we design the Cascade Attention Mask. This mechanism severed attention pathways between token masks from different instances during a single forward pass, while preserving their visibility to image and textual context. This design enables WOW-Seg to process multiple masks concurrently within a single forward pass without interference. This constitutes a key algorithmic contribution distinct from simple "serial classification" or "naive batching". Experiments demonstrate that Cascade Attention Mask achieves a 2.25 gain on the LVIS dataset.
>
> 3. **Dedicated Visual Prompt Design: Mask2Token**
>
> Mask2Token is not a simple clipping or background blurring technique. It is specifically engineered to map binary masks onto the pre-trained semantic space of VLLMs. Our ablation experiments (Table 6 & Fig. 7) demonstrate that Mask2Token outperforms simple mask clipping or blurring strategies in both convergence speed and final performance. This indicates it constitutes a carefully engineered algorithmic component for aligning VLLM feature spaces, rather than mere data preprocessing. Additionally, we present efficiency analyses for WOW-Seg. Compared to alternative approaches, WOW-Seg demonstrates significant speed advantages. Relevant analyses are provided in the anonymised link (https://anonymous.4open.science/r/3-7721).
>
> Consequently, we contend that the algorithmic innovations presented herein (addressing interference and efficiency challenges) are equally significant and intrinsically coupled with the dataset contribution.
>
> ---
>
> ## **Regarding the release of datasets.**
>
> We shall **certainly** release the RR-7K dataset. We firmly believe that RR-7K's greatest value as a large-scale open-world region recognition dataset lies in **driving collective progress within the community**. To ensure **the reproducibility** of our work and **facilitate future related research**, we shall **publicly release the complete RR-7K dataset, alongside our model code and weights**.
>
> Public links to all resources will be included in the final camera-ready version of our paper.

---

> > ### Comment · Reviewer_GZHt · 2025-11-27
> >
> > Thanks for their clarifications. I have no further concerns.

---

> > > ### Author Response · Authors · 2025-11-27
> > > **Authors' Appreciation to Reviewer GZHt**
> > >
> > > Once again, we extend our sincere gratitude for your affirmation and recognition of WOW-Seg, and we are also deeply appreciative of the valuable feedback you have provided. This input offers crucial guidance for refining our research.

---

### Official Review · Reviewer_7ZmL · 2025-10-30

**Soundness:** 3
**Presentation:** 3
**Contribution:** 3
**Rating:** 6
**Confidence:** 4

**Summary:**

This paper introduces WOW-Seg, a Word-free Open World Segmentation model designed for robust segmentation and semantic understanding across open-set object categories. It incorporates Mask2Token, which converts image masks into visual tokens aligned with the VLLM feature space, and Cascade Attention Mask, which reduces inter-instance interference. The authors also propose RR-7K, a large-scale region recognition benchmark with 7,662 categories. WOW-Seg achieves state-of-the-art performance on LVIS while using only one-eighth of the parameters, demonstrating strong open-world generalization and efficiency.

**Strengths:**

**Originality:** The paper introduces two novel and effective components. The first is Mask2Token, a visual prompt module that transforms masks into tokens, which cleverly embeds mask features directly into the VLLM's existing feature space. The second is the Cascade Attention Mask , a custom attention mechanism designed to solve inter-instance interference. This allows for efficient multi-mask processing by explicitly decoupling the features of different masks and their corresponding text outputs during autoregressive generation.

**Significance:** WOW-Seg achieves state-of-the-art results, surpassing prior models on LVIS (e.g., +4.1 Semantic IoU) and PACO , while being substantially more parameter-efficient (1B parameters vs. 3B-8B for SOTA competitors). Furthermore, the introduction of the RR-7K dataset is a major contribution, providing a much-needed benchmark to evaluate true open-world generalization, on which prior SOTA models perform poorly.

**Clarity:** The paper is well-written and easy to follow.

**Weaknesses:**

- Dependency on Mask Generator Quality: The model's "word-free" recognition is fundamentally dependent on the quality of the masks provided by an external "Mask Generator" (like SAM) during inference.
- The Mask2Token module, while effective, appears computationally intensive. It requires cropping each mask region and feeding it through the visual encoder to extract tokens. The computational cost and inference speed are not analyzed.
- Several closely related works [1,2,3,4] are missing from the discussion. The authors should provide a more in-depth analysis of these studies and incorporate them into the comparative evaluation to ensure a comprehensive and fair comparison.
- The paper lacks a quality analysis of the newly proposed RR-7K benchmark.

[1] Kawano, Yasufumi, and Yoshimitsu Aoki. "Tag: Guidance-free open-vocabulary semantic segmentation." arXiv preprint arXiv:2403.11197 (2024).

[2] Ülger, Osman, Maksymilian Kulicki, Yuki Asano, and Martin R. Oswald. "Auto-vocabulary semantic segmentation." In Proceedings of the IEEE/CVF International Conference on Computer Vision, pp. 24266-24275. 2025.

[3] Rewatbowornwong, Pitchaporn, Nattanat Chatthee, Ekapol Chuangsuwanich, and Supasorn Suwajanakorn. "Zero-guidance segmentation using zero segment labels." In Proceedings of the IEEE/CVF International Conference on Computer Vision, pp. 1162-1172. 2023.

[4] Shin, Heeseong, Chaehyun Kim, Sunghwan Hong, Seokju Cho, Anurag Arnab, Paul Hongsuck Seo, and Seungryong Kim. "Towards open-vocabulary semantic segmentation without semantic labels." Advances in Neural Information Processing Systems 37 (2024): 9153-9177.

**Questions:**

- PAM was trained on a large-scale dataset, yet it performs poorly on the RR-7K dataset. In contrast, WOW-Seg, which was trained only on a combination of LVIS, PACO, and COCO-Stuff datasets, achieves significantly higher performance on RR-7K. The authors are encouraged to provide an analysis explaining this discrepancy.
- In the open-vocabulary segmentation results presented in Table 2, it would be beneficial to include comparisons with more recent state-of-the-art methods, such as CAT-Seg, to provide a stronger benchmark evaluation.
- The proposed Cascade Attention Mask is designed to ensure that object masks are independent of each other. However, the results indicate that the multiple-masks-per-sample (MM) strategy outperforms the single-mask-per-sample (SM) strategy. The authors should clarify why MM remains effective if the masks are intended to be independent.

---

> ### Author Response · Authors · 2025-11-22
> **Authors' Response to Reviewer 7ZmL - Part 1/4**
>
> Thank you for your detailed and constructive feedback. We appreciate your recognition of WOW-Seg's innovative and efficient nature, particularly when compared to existing models.
>
> ---
>
> ## **Regarding the dependency on the mask generator.**
>
> To **thoroughly validate WOW-Seg's performance when confronted with imperfect, even extremely poor-quality visual prompts, we have supplemented a set of** comprehensive ablation experiments.
>
> We not only tested high-quality **SAM**-generated masks but also **further stress-tested the model** by using only **very coarse rotated bounding boxes** and **minimal bounding ellipses** as mask inputs. These rudimentary prompts provided only the object's approximate position and outline, falling far short of a refined segmentation mask.
>
> The experimental results are presented in the table below.
>
> **Table: Robustness analysis on different visual prompt sources.** RBB denotes the use of a rotating bounding box as a prompt mask. Ellipse denotes the use of a bounding ellipse as a prompt mask.
>
> | Mask Source                | LVIS (S. Sim.) | LVIS (S. IoU) | PACO (S. Sim.) | PACO (S. IoU) |
> | :------------------------- | :------------: | :-----------: | :------------: | :-----------: |
> | ***Geometric Heuristics*** |                |               |                |               |
> | GT Mask                    |      89.7      |     82.4      |      88.5      |     79.2      |
> | RBB                        |      87.3      |     78.6      |      85.5      |     74.1      |
> | Ellipse                    |      86.9      |     78.0      |      84.8      |     73.4      |
> | ***Mask Generator***       |                |               |                |               |
> | SAM (Huge)                 |      88.8      |     81.0      |      86.6      |     75.7      |
> | HQ-SAM (Huge)              |      88.9      |     81.1      |      86.2      |     75.3      |
> | SAM2 (Large)               |      89.0      |     81.2      |      86.8      |     76.3      |
>
> As can be clearly seen from the table above, when using general-purpose mask generators such as SAM, performance only experienced a slight decline compared to using gt masks. This demonstrates the practical efficacy of WOW-Seg. However, the most surprising finding is that even when fed extremely crude rotated bounding boxes as input, WOW-Seg maintains remarkable recognition capability (achieving an S. Sim. of 87.3 on LVIS).
>
> In summary, this comprehensive set of experiments demonstrates that WOW-Seg's exceptional recognition capabilities **are not confined to ideal conditions**. Our model exhibits **remarkable robustness** to the quality of upstream visual prompts, effectively processing everything from high-quality GT Masks to extremely crude elliptical masks.
>
> ---
>
> ## **Regarding the computational cost of Mask2Token.**
>
> Your concern ("Mask2Token appears computationally intensive") is a perfectly logical intuition. We wish to clarify a crucial point here, substantiated by the computational cost analysis you suggested (and which we have now supplemented): **WOW-Seg is not only computationally non-intensive, but in multi-mask inference, it is significantly more computationally efficient than prior works such as PAM.**
>
> 1. Architectural Comparison.
>
>    PAM is a **single-mask inference** model. It can only process one mask at a time. Therefore, to process an image containing N objects, PAM must **execute N complete forward passes sequentially**. Its computational cost (time and FLOPs) scales **linearly** with the number of masks N.
>
>    **WOW-Seg (Ours):** Our model (leveraging Mask2Token and Cascade Attention Mask) is designed for **parallel multi-mask inference**. We can process N masks simultaneously within a single forward pass. Its computational cost is not linearly proportional to the number of masks N.
>
> 2. Additional experimental evidence: computational cost analysis
>
> To substantiate this with data, we conducted the computational cost analysis you requested. We compared the total inference time and FLOPs for PAM and WOW-Seg as the number of masks increased from 1 to 32.
>
> As illustrated, WOW-Seg consistently outperforms PAM in forward efficiency, whether employing a single mask or multiple masks. Most notably, increasing the number of masks from 1 to 32 (a 32-fold increase) resulted in WOW-Seg's total FLOPs and inference time increasing by less than fourfold. (The experimental result graph is in the anonymous link: https://anonymous.4open.science/r/2-0B67/)

---

> ### Author Response · Authors · 2025-11-22
> **Authors' Response to Reviewer 7ZmL - Part 2/4**
>
> 3. Why is WOW-Seg so efficient?
>
>    Your intuition might suggest that "passing through ViT" is costly. However, as our experiments confirm, within the VLLM architecture, the **computational cost of the ViT encoder** is not the primary factor when compared to that of the **large language model (LLM) decoder**.
>
>    **The ingenuity of Mask2Token:** Our Mask2Token module (as shown in Figure 3) efficiently distils each mask (regardless of size) into **a minimal number of representative tokens** (approximately 20 on average).
>
>    **Cost Analysis:**
>
>    - **ViT Cost:** Although we run N parallel ViT pruning operations for N masks, GPUs handle this batch processing extremely efficiently. This incremental cost is negligible.
>    - **LLM Cost:** **The true computational bottleneck lies in the LLM's self-attention mechanism**. PAM must run N times, whereas WOW-Seg **runs only once**.
>    - We merely increase the LLM's input sequence length from `L` (base tokens) to `L + (N * 20)`. Within the total VLLM computation, this increment (even for N=32) is negligible.
>
> Thank you for raising this valuable point. It has prompted us to quantify a key advantage of WOW-Seg. Our analysis demonstrates that Mask2Token is not only not computationally intensive, but is in fact pivotal to achieving **highly efficient parallel multi-mask inference**. We shall incorporate **this computational cost analysis diagram and the aforementioned discussion** into our final paper to fully substantiate the efficiency advantages of our approach.
>
> ---
>
> ## **Regarding the relevant work [1, 2, 3, 4].**
>
> We are most grateful to the reviewers for recommending these insightful works [1-4]. We have carefully studied these papers and fully concur that they represent a highly significant research direction within the field of open-world understanding. **We shall cite and discuss these works in depth within the revised manuscript.**
> However, regarding the suggested "comparative evaluation", we wish to clarify the fundamental difference in task paradigms between WOW-Seg and these methods. This disparity renders direct quantitative comparisons (e.g., on RR-7K or LVIS) technically unfeasible and unfair to both parties.
>
>
> - **Regarding References [1-4]:** These approaches aim to address both "find region" and "semantic annotation" *simultaneously*. They typically employ a bottom-up, feature-based approach to automatically extract masks from images. Their core challenge lies in "how to find region without supervision".
> - **Regarding WOW-Seg:** As defined in our introduction and methodology sections, WOW-Seg follows the "region recognition" paradigm (consistent with works such as Osprey, DAM, PAM, etc.). Our approach is prompt-dependent: the model receives a specific mask (provided by the user or pre-generated by generators like SAM) as input and focuses on leveraging the semantic space of the VLLM to accurately **identify** that region. Our core task is "given a mask, determine what it is," rather than "discovering the mask from scratch."
> - Our test benchmarks (RR-7K, LVIS) evaluate a model's recognition accuracy for a given mask. The cited works [1-4] are end-to-end segmentation models incapable of receiving external mask prompts for specific classification tasks; they can only output regions discovered autonomously. This precludes calculating Semantic Similarity or Semantic IoU based on specific masks.
>
> [1] Kawano, Yasufumi, and Yoshimitsu Aoki. "Tag: Guidance-free open-vocabulary semantic segmentation." arXiv preprint arXiv:2403.11197 (2024).
>
> [2] Ülger, Osman, Maksymilian Kulicki, Yuki Asano, and Martin R. Oswald. "Auto-vocabulary semantic segmentation." In Proceedings of the IEEE/CVF International Conference on Computer Vision, pp. 24266-24275. 2025.
>
> [3] Rewatbowornwong, Pitchaporn, Nattanat Chatthee, Ekapol Chuangsuwanich, and Supasorn Suwajanakorn. "Zero-guidance segmentation using zero segment labels." In Proceedings of the IEEE/CVF International Conference on Computer Vision, pp. 1162-1172. 2023.
>
> [4] Shin, Heeseong, Chaehyun Kim, Sunghwan Hong, Seokju Cho, Anurag Arnab, Paul Hongsuck Seo, and Seungryong Kim. "Towards open-vocabulary semantic segmentation without semantic labels." Advances in Neural Information Processing Systems 37 (2024): 9153-9177.
>
> ----
>
> ## **Analysis of the quality of RR-7K.**
>
> Your opinion is of great importance. We will conduct a more comprehensive quality analysis of the RR-7K. Since the reply contains pictures, we have placed the presentation in an anonymous link.
>
> Anonymous link: https://anonymous.4open.science/r/2-0B67/
>
> If you encounter garbled characters while reading online, please **download** the content for viewing. Thank you for your understanding.
>
> ----

---

> ### Author Response · Authors · 2025-11-22
> **Authors' Response to Reviewer 7ZmL - Part 3/4**
>
> ## **Regarding the performance of PAM.**
>
> We are grateful to the reviewer for raising this insightful question. The fact that PAM performs less favourably than WOW-Seg on the RR-7K dataset does indeed appear counterintuitive. However, upon thorough analysis, we have identified two key factors primarily responsible for this discrepancy: **differences in core architecture** and **bias in the training data distribution**.
>
>
> 1. Core architecture differs.
>
>    As demonstrated in our paper (WOW-Seg) in Figure 1(c) and Appendix Table 7, the most significant distinction between RR-7K and LVIS/PACO lies in its inclusion of **a vast array of ‘Stuff’ classes and environmental categories** (such as "field", "canyon", "ocean wave"). These objects are typically very large, occupying the majority of the image area.
>
>    The PAM architecture faces fundamental difficulties in processing such objects:
>
>    - **PAM's feature source:** PAM relies on SAM2 as its visual backbone. The SAM series of models are "Instance-Centric" segmenters. They are designed to segment "Things" (e.g., "person", "car"), not ‘Stuff’ (e.g., "sky", "grass"). When used alone, both SAM and SAM2 exhibit extremely poor performance in segmenting stuff objects [5].
>    - **Limitations of input prompts:** PAM frequently employs **BBoxes (bounding boxes) as prompts during training and inference. When a large BBox is used to prompt a "Stuff" class (e.g., a BBox encompassing most of the sky), SAM's "instance" features become confused and meaningless (as buildings and trees are also enclosed within the same BBox)**. It fails to provide coherent, discriminative features for this region because it attempts to find "Things" within "Stuff".
>    - **Osprey's corroboration:** Osprey's significantly superior performance over PAM on RR-7K validates this argument. Osprey accepts **masks** as input and employs RoI operations to extract features directly from the masked regions within feature maps. As observed, this mask-based operation effectively **filters out background noise within the bounding box**, enabling the model to focus on the actual pixels of irregular objects. This explains why Osprey outperforms PAM on the highly challenging RR-7K dataset, which contains a substantial number of Stuff classes.
>
>    In contrast, WOW-Seg possesses an architectural advantage.
>
>    - Our **Mask2Token** module fundamentally addresses this issue. We **do not** rely on a fixed, global, "Instance-Centric" feature map.
>    - Instead, we first **crop** the region containing the mask (with context) before feeding it into our Vision Encoder. This ultimately represents the current mask feature with only around 20 tokens, ensuring both performance and efficiency.
>    - This "**Zoom-in**" mechanism ensures the model generates **high-fidelity, high-resolution, region-specific** visual tokens for both small objects ("cup") and vast environmental areas ("field"). This grants WOW-Seg an overwhelming advantage in identifying the numerous "Stuff" classes within RR-7K.
>
> 2. **Biases in the Training Data Distribution**
>
>    **The "Scale" Illusion of Training Data:** Although PAM utilises large-scale datasets, we must scrutinise the **nature** of its data.
>
>    - **Descriptive Depth vs. Recognition Breadth:** PAM's primary advantage in large-scale data lies in **fine-grained description** and **understanding relationships between objects**. Such data can teach models to generate complex sentences (e.g., "a cat lying beside the sofa"), but this does not equate to expanding the model's **capacity for recognising atomic categories**. Extensive descriptive training does not directly translate into precise classification capabilities for the 7,000+ specific categories in RR-7K (particularly long-tail vocabulary).
>
> 3. PAM's failure is not coincidental but stems from the combined effects of **data distribution** and **architectural constraints**.  This further underscores RR-7K's necessity as an evaluation benchmark for assessing model performance in complex, open-world scenarios rich in Stuff classes.
>
> [5] Ke, Lei, et al. "Segment anything in high quality." *Advances in Neural Information Processing Systems* 36 (2023): 29914-29934.
>
> ---

---

> ### Author Response · Authors · 2025-11-22
> **Authors' Response to Reviewer 7ZmL - Part 4/4**
>
> ## **Regarding comparison with CAT-Seg.**
>
> We appreciate the reviewers' suggestions. We fully acknowledge that CAT-Seg is a robust state-of-the-art method in this field. Regarding why CAT-Seg was not included in Table 2 but compared in Table 3, we wish to clarify our experimental design rationale.
>
> 1. The experimental setup in Table 2 is unsuitable for comparing CAT-Seg.
>
>    - **Setting consistency:** As noted in Table 2's caption, this experiment utilises **Ground Truth (GT) Mask/Box** as input. The competing methods listed (e.g., Shikra, Osprey, GPT4ROI) are multimodal models capable of receiving specific visual prompts and outputting semantic categories.
>
>    - **Methodological Incompatibility:** CAT-Seg is an end-to-end semantic segmentation framework designed to process entire images and output segmentation maps. It lacks an interactive interface capable of "receiving a specific mask and classifying only that mask".
>
> 2. We have conducted a fair comparison with CAT-Seg in Table 3.
>
>    In response to the reviewer's request for "stronger benchmark evaluation," we have in fact **already conducted a direct comparison with CAT-Seg in Table 3**. In Table 3, we employ the metric of **mask classification accuracy**. This approach, utilising the MaskCLIP++ framework, ensures CAT-Seg and WOW-Seg process identical ground truth regions. This isolates the impact of segmentation quality, enabling a pure comparison of their respective capabilities in classifying/recognising regional features.
>
> ----
>
> ## **Regarding why Cascade Attention Mask achieves high performance under MM.**
>
> We are grateful for raising it, as it affords us the opportunity to clarify a crucial point: the phenomenon you observed (MM > SM) is precisely the motivation for introducing the Cascade Attention Mask, rather than contradicting it.
>
> The results in Table 4 do indeed reveal two distinct yet concurrent phenomena:
>
> - **Data Efficiency:** MM vs. SM
> - **Feature Interference:** MM vs. MM (with Cascade Attention Mask)
>
> Please allow us to break this down in detail:
>
> 1. Why is MM (Row 2) > SM (Row 1)? **(Data Efficiency)**
>
>    Your observation that ‘the MM strategy outperforms the SM strategy’ is **entirely correct**. As shown in **Table 4**, the performance of SM (LVIS S. IoU 74.70) falls significantly below that of the standard MM (LVIS S. IoU 80.10).
>
>    **Reason:** Due to **training efficiency**.
>
>    **Evidence:** As stated in Section S4.5 of the paper, "all experiments ensured identical training steps". This implies that during the same 10,000-step training, the SM model observed only 10,000 mask instances, whereas the MM model (assuming an average of 10 masks per image) observed 100,000 mask instances.
>
>    **Conclusion 1:** The MM strategy learns **more data** within the same training duration, consequently achieving significantly superior performance.
>
> 2. Where does the issue lie with MM (Row 2)? **(Feature Interference)**
>
>    However, whilst MM (Row 2) excels in data efficiency, it carries an inherent **limitation that constrains its performance**: **"inter-instance interference"**.
>
>    - **Issue:** During standard MM training (without Cascade Attention Mask), the model's attention "leaks" when predicting labels for "Mask A", inadvertently perceiving features from "Mask B", "Mask C", etc. This creates a confusing training signal, leaving the model uncertain about which instance it should generate labels for.
>
>    - **Evidence:** This explains why we introduced the Cascade Attention Mask in our S4.5 ablation experiments.
>
> 3. Why is MM+Cascade Attention Mask (Row 5) > MM (Row 2)? **(The Effectiveness of Cascade Attention Mask)**
>
>    This constitutes the **true** ablation experiment for evaluating the value of Cascade Attention Mask.
>
>    - **Comparison:** We must compare **standard MM** (Row 2, LVIS S. IoU 80.10) with **MM employing our Cascade Attention Mask** (Row 5, LVIS S. IoU 82.35).
>
>    - **Conclusion 2:** Under **controlled data efficiency** (both being MM), adding CAM yields a **+2.25** (S. IoU) significant performance gain.
>
>    - **Demonstration:** This +2.25 gain **directly quantifies** the **severity** of "inter-instance interference" suffered by the standard MM, and **irrefutably demonstrates** the efficacy of our Cascade Attention Mask in addressing this issue.

---

> > ### Comment · Reviewer_7ZmL · 2025-11-26
> >
> > I thank the authors for their thorough response, and I am pleased to maintain my positive rating.

---

> > > ### Author Response · Authors · 2025-11-26
> > > **Authors' Appreciation to Reviewer 7ZmL**
> > >
> > > Thank you for your positive feedback and ongoing support. We sincerely appreciate the time and care you have dedicated to reviewing our work.

---

### Official Review · Reviewer_AFoP · 2025-11-01

**Soundness:** 3
**Presentation:** 2
**Contribution:** 2
**Rating:** 4
**Confidence:** 4

**Summary:**

This article focuses on the problem of open world image segmentation. It introduces a word-free open world segmentation model for segmenting and recognizing objects from open set categories. The core idea is to encode multiple masks as visual prompts and autoregressively identifies each mask. It develops Mask2Token for mask encoding and  Cascade Attention Mask for detection. Further, the article introduces a new dataset, RR-7K, which features with 7000+ categories.

**Strengths:**

- The work contributes a new dataset, RR-7K, which holds notable potential to benefit the research community.
- The proposed method achieves state-of-the-art performance across multiple benchmarks

**Weaknesses:**

- The motivation behind the cascade attention mechanism is not sufficiently justified or explained. While the authors acknowledge that “in the natural world, there are inherent correlations between objects,” they ignore this prior knowledge in their design — which seems counterintuitive.

- Furthermore, the proposed cascade attention scheme lacks a formal mathematical description. The current exposition relies heavily on intuition.

- For the dataset, I am unsure whether we indeed need to recognize 7K categories. As observed in Table 7, many categories appear semantically overlapping or synonymous (e.g., "audience" vs. "audience member", "ad board" vs. "advertisement board"). This suggests potential redundancy that could inflate category counts without adding meaningful discriminative power. Moreover, the paper provides no statistical analysis of the dataset, such as, label distribution.

- The current visualizations focus primarily on specific objects, however, it is expected to show the overall segmentation results of entire images.

- As the leading information that authors intend to share, the caption of Fig. 1 is minimal and lacks sufficient context. TBH, it is not easy to grasp all details at the first glance.

**Questions:**

- Why authors disregard the inherent semantic relationships among objects when designing the cascade attention mask?

- Could you provide details on the label distribution across the categories in the RR-7K dataset?

- While InternVL-78B is employed to filter hallucinations during data annotation, how can you ensure that the model itself does not introduce hallucinations?

- Osprey exhibits poor performance on established benchmarks like LVIS and PACO, yet achieves relatively good results on the proposed RR-7K dataset. What explains this discrepancy? A thorough investigation into this is essential to validate RR-7K as a fair, representative, and necessary benchmark.

---

> ### Author Response · Authors · 2025-11-22
> **Authors' Response to Reviewer AFoP - Part 1/4**
>
> Thank you for your valuable feedback, which highlights the strengths of our work, particularly the introduction of the RR-7K dataset and the performance of our method in benchmark testing.
>
> ---
>
> ## **Motivation and Mathematical Description of the Cascade Attention Mask.**
>
>
> We sincerely thank the reviewers for pointing this out. In fact, we have not overlooked semantic relevance. In fact, the design of Cascade Attention Mask is **not intended to ignore beneficial semantic correlations**, but rather to **address a specific technical challenge: inter-instance feature interference**. During multi-mask (MM) training, this interference causes the model to experience attribution confusion, where it becomes unclear which masked instance corresponds to a given predicted label.
>
> ### **Clarification: A more specific and apt example**
>
> Please allow us to illustrate with a clearer example what harmful correlations or interference specifically entail, and why the Cascade Attention Mask is required.
>
> 1. **Scenario:** Suppose an image (as shown in Figure 1(a) or Figure 2) contains two distinct chef instances, which we refer to as **Chef A** and **Chef B**.
>
>      - **Input Sequence:** The token sequence received by the model is approximately as follows: `[Image Tokens] [Text "Please segment all masks"] [Mask_A Tokens (Chef A)] [Mask_B Tokens (Chef B)]`
>      - **Target Output:** The model must autoregressively generate: `[chef], [chef]`
>
> 2. **The crux of the matter: Standard multi-mask (MM) training without a Cascade Attention Mask**
>
>    In standard autoregressive models (such as the Causal Attention Mask in Figure 4(a)), when predicting the **first** `[chef]` token (corresponding to Chef A), the model's attention **can simultaneously perceive** both `[Mask_A Tokens]` and `[Mask_B Tokens]`.
>
>    - **Harmful interference:** This generates a **confusing training signal**. The model is required to output a label while simultaneously receiving visual features from two distinct instances (A and B). It cannot determine whether the `[chef]` label should be attributed to `Mask_A` or `Mask_B`.
>    - **Information leakage:** This constitutes what we term "harmful information leakage" or "inter-instance interference". Features from `Mask_B` leak into the prediction process for `Mask_A`.
>
> 3. **Solution: Employing Cascade Attention Mask (our contribution)**
>
>    Cascade Attention Mask (as illustrated in Figure 4(b)) achieves this by modifying the attention matrix to **force decoupling** of instance attribution.
>
>    **How Cascade Attention Mask operates:**
>
>    - **Independent Masking:** The Cascade Attention Mask ensures `[Mask_A Tokens]` and `[Mask_B Tokens]` remain invisible to each other.
>
>    - **1-to-1 Mapping Between Output and Masks:**
>
>      When the model predicts the **first** `[chef]` (corresponding to A), the Cascade Attention Mask **only permits** it to attend to `[image tokens]`, `[text]`, and `[Mask_A tokens]`. It is **explicitly prohibited** from seeing `[Mask_B tokens]`.
>
>      Similarly, when the model predicts the **second** `[chef]` (corresponding to B), it is **only permitted** to attend to `[image tokens]`, `[text]`, and `[Mask_B tokens]`, while being unable to see `[Mask_A tokens]`.
>
>
> ### **Clarification: We Have Not Overlooked “Useful” Relevance**
>
> Reviewers expressed concern that we might have disregarded useful contextual elements such as "cup and water". **This is not the case.**
>
> - **Beneficial Context (Global Semantics):** When predicting `[Mask_A Tokens]` (Chef A), the model still retains access to all `[Image Tokens]`. This means the model can perceive the entire kitchen scene. This global context (e.g., perceiving stove and pot) is **beneficial** for correctly identifying "chef", and our Cascade Attention Mask **preserves** this beneficial context.
> - **Harmful interference (instance features):** The Cascade Attention Mask **only blocks** feature interference from **other specific instances** (such as `Mask_B Tokens`), as these features would confuse the model's understanding of its current task (i.e., "describe specifically Mask_A").

---

> > ### Author Response · Authors · 2025-11-22
> > **Authors' Response to Reviewer AFoP - Part 2/4**
> >
> > ### **Empirical Evidence: Why the Cascade Attention Mask is Crucial**
> >
> > Our paper's **ablation experiments (Table 4)** provide compelling empirical support for this rationale:
> >
> > 1. **MM vs SM:** Performance using only Multi-Mask (MM) without the Cascade Attention Mask (Table 4, Row 2) (LVIS Sem. IoU 80.10) significantly outperforms Single-Mask (SM) training (Table 4, Row 1, Sem. IoU 74.70). This indicates MM training (processing multiple objects simultaneously) is more efficient and learns from a richer dataset.
> > 2. **MM+Cascade Attention Mask vs MM:** Incorporating the Cascade Attention Mask (Table 4, Row 5) yields a **further substantial improvement** to 82.35 (Semantic IoU).
> > 3. **Conclusion:** The **+2.25 point** improvement from MM (80.10) to MM+Cascade Attention Mask (82.35) **directly quantifies the detrimental impact of inter-instance interference**. This demonstrates that interference is a genuine phenomenon, and that Cascade Attention Mask successfully mitigates it, yielding substantial performance gains.
> >
> > ### **Mathematical description**
> >
> > Thank you for highlighting this crucial point. To better present the Cascade Attention Mask, we have added its mathematical description as follows.
> >
> > This is a portion of the original text providing a mathematical description for the case where **the Cascade Attention Mask is not employed**:
> > With the effect of the original causal attention mask of the large language model, assuming there are K objects, the probability of these K object names being predicted is shown in the following formula:
> > $$
> > P\left(O_1, O_2, \ldots, O_K \mid Image ; T ; M\right)=\prod_{i=1}^K P\left(O_i \mid Image ; T ; M ; O_0, O_1, \ldots, O_{i-1}\right).
> > $$
> >
> > Among them, $O$ represents the object to be predicted, $Image$ represents the image token, $T$ represents the text token, and $M = \{m_1, m_2,\ldots,m_K\}$ denotes the set of all object mask tokens. It is easy to observe that when predicting the $i$-th object, the model refers to the mask prompts and output results from the $0$-th to the $(i-1)$-th objects. We hope that the name of the $i$-th object is guided solely by its corresponding $i$-th mask.
> >
> > This is a newly added paragraph to describe the situation when **using the Cascade Attention Mask**:
> > With the Cascade Attention Mask, the predictions for different objects are conditionally independent. Specifically,
> > $$
> > P\left(O_1, O_2, \ldots, O_K \mid Image ; T ; M\right) = \prod_{i=1}^K P\left(O_i \mid Image ; T ; M \right),
> > $$
> >
> > where $M = \{m_1, m_2,\ldots,m_K\}$ denotes the set of all object mask tokens. This factorization reflects the fact that,  with the Cascade Attention Mask, the prediction of the $i$-th object $O_i$ is conditionally independent of the mask tokens of all other objects given the image and text tokens.
> >
> > Although the conditioning set $M$ includes the mask tokens of all objects, the attention pattern enforces that $O_i$ can only attend to its own mask tokens $m_i$. As a result, the probability term satisfies:
> >
> > $$
> > P\left(O_i \mid Image ; T ; M \right) = P\left(O_i \mid Image ; T ; m_i \right),
> > $$
> >
> > meaning that only the $i$-th mask contributes to the prediction of the $i$-th object.
> >
> > ### Summary
> >
> > The motivation for the Cascade Attention Mask is to **address instance attribution confusion in multi-mask training**, not to disregard beneficial semantic context. We will revise Section S3.3 in the final version of the paper to accurately articulate the necessity and compelling justification for the Cascade Attention Mask.
> >
> > ----
> >
> > ## **Distribution of RR-7K, Figure 1 Caption, and Entire Visualization of Image Results**
> >
> > Since all three of these questions involve the display of pictures, for an intuitive presentation, we provide the anonymous link.
> >
> > In the anonymous link, we present a detailed visualization of the RR-7K distribution, along with the revised Figure 1 caption and entire results of several images.
> >
> > Anonymous link: https://anonymous.4open.science/r/1-DC67
> >
> > If you encounter garbled characters while reading online, please **download** the content for viewing. Thank you for your understanding.
> >
> > ----

---

> > > ### Author Response · Authors · 2025-11-22
> > > **Authors' Response to Reviewer AFoP - Part 3/4**
> > >
> > > ## **Regarding the Illusion of InternVL-78B**
> > >
> > > We wish to clarify that InternVL-78B may exhibit some hallucinations. However, under our designed data annotation workflow, these hallucinations will not impact the accuracy of the final dataset. In our data annotation process, InternVL-78B is not the final arbiter of quality—it is human review that serves as the ultimate guarantee.
> > >
> > > 1. The role of InternVL-78B: a high-recall "verifier", not a "generator"
> > >
> > >    First, we have significantly reduced the risk of hallucinations in InternVL-78B by shifting its task from "open-ended generation" to "**closed-verification**". This approach is grounded in a principle extensively validated within academia: **large models demonstrate far greater robustness and reduced susceptibility to hallucinations when performing "closed-verification" tasks (such as "yes/no" judgements) compared to executing "open-ended generation" tasks** [1, 2].
> > >
> > >    - **Non-hallucinatory task:** We do not require it to generate labels (which readily leads to hallucinations).
> > >
> > >    - **Verification task:** We pose the question: "Is this red masked region [category name]? Please respond with 'yes' or 'no'."
> > >
> > >    This binary classification task of "yes/no" proves far more robust and reliable than open-ended generation.
> > >
> > > 2. Two Potential Error Types and Their Handling
> > >
> > >    **Case 1: False Negative – InternVL-78B hallucinates and answers "No"**. Should InternVL-78B erroneously reject a correctly labelled instance, the data will be discarded.
> > >
> > >    **Impact:** Such errors **do not compromise the final dataset's accuracy. They merely slightly reduce the dataset's recall rate, resulting in the loss of some potentially included samples.**
> > >
> > >    **Case 2: False Positive - InternVL-78B Hallucinates and Answers "Yes"**. For instance, QwenVL might erroneously label a "fire hydrant" as a "robot", while InternVL-78B hallucinates and falsely validates "Yes, this is a 'robot'".
> > >
> > >    **Impact:** This error **will** introduce hallucinated labels into our dataset.
> > >
> > > 3. Our Solution: Manual Screening
> > >
> > >    We fully anticipated the risks associated with Case 2. This is precisely why our process does not conclude with VLLM filtering.
> > >
> > >    As illustrated in **Figure 5**, our procedure incorporates a third and most critical stage: "Manual Screening".
> > >
> > >    We organised a **50-person professional data annotation team** to conduct **manual re-examination** of **all** data passing InternVL-78B filtering (i.e., all samples where the VLLM answered ‘yes’).
> > >
> > >    **Process:** Human annotators view the image, mask, and VLLM-confirmed label. Their task is to identify and remove all "false positive" hallucination samples resulting from errors made by both VLLMs (including QwenVL and InternVL).
> > >
> > > 4. **Summary:** InternVL-78B serves solely as an **automated pre-filter** designed to substantially reduce the volume of data requiring human review (i.e., filtering out numerous obvious ‘no’ samples).
> > >
> > >    The final clean state of our RR-7K dataset is **not guaranteed by InternVL-78B, but rather by manual screening**.
> > >
> > > ----

---

> > > > ### Author Response · Authors · 2025-11-22
> > > > **Authors' Response to Reviewer AFoP - Part 4/4**
> > > >
> > > > ## **Regarding the performance differences of the Osprey on the LVIS and RR-7K.**
> > > >
> > > > We greatly appreciate you for raising this point, as it directly addresses the **core value** of RR-7K as a new benchmark.
> > > >
> > > > Osprey underperforms on LVIS yet outperforms PAM on RR-7K, and this performance discrepancy is not a flaw or unfairness of the RR-7K benchmark but a logically expected outcome. It precisely **demonstrates** the **limitations** of existing benchmarks like LVIS in evaluating "open-world" capabilities, and highlights the **necessity** of RR-7K as a fairer and more representative benchmark.
> > > >
> > > > This phenomenon can be attributed to two key factors: output format and input modality.
> > > >
> > > > ### **Osprey's suboptimal performance on LVIS (output format)**
> > > >
> > > > Osprey performs poorly on LVIS/PACO. This is largely attributable to **format mismatch between model outputs and ground truth**.
> > > >
> > > > - **Osprey's Output:** Osprey is trained to generate **descriptive phrases** for regions (e.g., "a white cat sitting on a mat").
> > > > - **LVIS/PACO Ground Truth:** The ground truth for these benchmarks is a **single categorical word** (e.g., "cat").
> > > > - **Metric Penalty:** Our primary metric is "Semantic Similarity". Whilst "a white cat..." is semantically correct, its similarity score to the GT "cat" will **never exceed 1.0**. In contrast, models like PAM, trained to output categorical terms directly, more readily achieve the full 1.0 score.
> > > >
> > > > Therefore, Osprey's "low score" on LVIS is to some extent a metric artefact.
> > > >
> > > > ### **Performance Reversal on RR-7K (input modality)**
> > > >
> > > > Why does Osprey outperform PAM on RR-7K? This reveals a **fundamental blind spot** in LVIS/PACO as benchmarks. This blind spot was "overfitted" by PAM but successfully exposed by RR-7K.
> > > >
> > > > **Osprey** is a mask-based model. It receives precise pixel-level masks as input and extracts features from that exact region.
> > > >
> > > > PAM, despite employing SAM2, explicitly states in its architecture that it takes **bounding boxes** as input, with SAM2 performing segmentation within these boxes.
> > > >
> > > > The vast majority of LVIS/PACO imagery consists of distinct ‘objects’ (Things). For objects, bounding boxes serve as a reasonable proxy.
> > > >
> > > > However, our RR-7K contains not only "objects" but also numerous "material/background" (Stuff) categories (e.g., "sky", "sand", "road").
> > > >
> > > > - For "Stuff" categories, **bounding boxes are an extremely poor, ambiguous input**. When using SAM or SAM2 in practice, handling stuff objects proves challenging. A bounding box encompassing "sky" inevitably includes substantial non-sky objects (e.g., buildings, trees) [3].
> > > > - This results in PAM receiving **chaotic** features, as it cannot discern which specific "Stuff" entity the user intends to query from SAM2's segmentation of a large bounding box.
> > > > - By contrast, Osprey receives a **precise mask**, enabling it to accurately comprehend the queried region. This confers a **fundamental architectural advantage** when handling the ‘Stuff’ category (WOW-Seg similarly benefits).
> > > >
> > > > In summary, this performance discrepancy is **entirely reasonable and crucially important**.
> > > >
> > > > 2. **PAM's Overfitting:** PAM's architecture (Box input) renders it **highly specialised** for benchmarks like LVIS, yet **unable to generalise** to a truly open world containing "Stuff".
> > > > 4. **Rationale for Performance Reversal:** On RR-7K, Osprey and WOW-Seg (Mask-based) naturally outperform PAM (Box-based).
> > > >
> > > > [1] Dhuliawala, Shehzaad, et al. "Chain-of-verification reduces hallucination in large language models." *Findings of the Association for Computational Linguistics: ACL 2024*. 2024.
> > > >
> > > > [2] Madaan, Aman, et al. "Self-refine: Iterative refinement with self-feedback." *Advances in Neural Information Processing Systems* 36 (2023): 46534-46594.
> > > >
> > > > [3] Ke, Lei, et al. "Segment anything in high quality." *Advances in Neural Information Processing Systems* 36 (2023): 29914-29934.

---

### Meta-Review · Area_Chair_26bJ · 2026-01-05

**Summary:**

The paper introduces WOW-Seg, a word-free open-world segmentation model that leverages a vision-language large model to perform region recognition from visual prompts without relying on predefined vocabularies. Reviewer AFoP and RZJ5 concern about cascade attention mechanism and lack a formal mathematical analysis. Additionally, most reviewers care about the statistical analysis in RR-7K. Some reviewers also care about the inference speed (7ZmL), entire visualization (AFoP), missing related works (7ZmL, RZJ5), limited novelty of the two-stage segmentation pipeline (GZHt) and the unfair comparison (RZJ5).

**Reviewer Concerns:**

The authors explain carefully their proposed cascade attention mechanism, but do not provide a satisfactory mathematical analysis. Initially, the authors show limited statistical analysis about RR-7K, but give a sufficient analysis in their response. For the concerns about inference speed, entire visualization, limited novelty, and unfair comparison, the authors provide detailed explanation in their response, which addresses most concerns of reviewers.

**Reviewer Scores:**

The authors have addressed most mainly concerns, but still need mathematical method to analyze why cascade attention mechanism works rather than heavily on intuition.

---

### Decision · Program_Chairs · 2026-01-26

Accept (Poster)